## PROCEEDINGS A

volcanology, mathematical modelling, computer modelling and simulation

Surtseyan eruption, magma bombs, magma fragmentation, flow in porous media

**Author for correspondence:**
Emma Greenbank
e-mail: emma.greenbank@gmail.com

# A theoretical model of Surtseyan bomb fragmentation

Emma Greenbank[1], Mark J. McGuinness[1] and C. Ian Schipper[2]

[1]School of Mathematics and Statistics, and [2]School of Geography, Environment and Earth Sciences, Victoria University of Wellington, Wellington, New Zealand

EG, 0000-0002-2332-3844

Surtseyan eruptions are an important class of mostly basaltic volcanic eruptions first identified in the 1960s, where erupting magma at an air–water interface interacts with large quantities of slurry, a mixture of previously ejected tephra that re-enters the crater together with water. During a Surtseyan eruption, hot magma bombs are ejected that initially contain pockets of slurry. Despite the formation of steam and anticipated subsequent high pressures inside these bombs, many survive to land without exploding. We seek to explain this by building and solving a simplified spherical mathematical model that describes the coupled evolution of pressure and temperature due to the flashing of liquid to vapour within a Surtseyan bomb while it is in flight. Analysis of the model provides a criterion for fragmentation of the bomb due to steam pressure build-up, and predicts that if diffusive steam flow through the porous bomb is sufficiently rapid the bomb will survive the flight intact. This criterion explicitly relates fragmentation to bomb properties, and describes how a Surtseyan bomb can survive in flight despite containing flashing liquid water, contributing to an ongoing discussion in volcanology about the origins of the inclusions found inside bombs.

## 1. Introduction

Surtseyan eruptions are characterized by significant bulk interactions of water with ascending magma [1–5], as

**Figure 1.** Cartoon of a Surtseyan eruption. Numbers refer to the water vapour cloud (1), ash in cypress tree shapes with bombs at their tips (2), crater (3), abundant water at crater level (4), layers of lava and ash (5), stratum (6), magma conduit (7), magma chamber (8), dike (9). Copyright: Creative Commons license, https://commons.wikimedia.org/wiki/File:Surtseyan_Eruption-numbers.svg, attributed to © Sèmhur/Wikimedia Commons/CC-BY-SA-3.0 (or Free Art License). (Online version in colour.)

exemplified by the volcano off the coast of Iceland that rose above the sea to become the island of Surtsey in the 1960s [1,6].

Like Hunga Ha'apai in Tonga in 2009 and Copelinhos in the Azores from 1957–1958, volcanoes that have risen under the sea and have a crater rim that is at sea level may exhibit this style of eruption. Mount Ruapehu in New Zealand with its crater lake sometimes erupts in this way.

Magma–water interaction plays an important role in determining volcanic eruption styles [3]. The style of interaction can vary greatly, from deep submarine scenarios, where quenching of magma causes relatively passive thermal granulation, to terrestrial scenarios, where rapid heat exchange between magma and ground or surface water can enhance magma fragmentation and explosivity [7]. Surtseyan eruptions, initially thought of as shallow submarine events, are now understood to involve interaction with any shallow standing water body, including lakes, rivers and marine waters [5].

Whatever the conditions under which magma and water meet, their interaction is limited in large part by the extreme difficulty of mixing fluids with such strongly contrasting viscosities and other physical properties [8]. Surtseyan eruptions, illustrated in figure 1, are unique, in that the water–magma interaction occurs in near-surface, periodically flooded vents, where there is re-entry of in-vent water-saturated slurry, a mixture of water and previously erupted ejecta that has fallen back into the crater [2,4]. Fresh magma that rises to encounter this slurry can more readily mix with it than with pure water, owing to the slurry's higher bulk viscosity and lower heat capacity [9]. The almost silent jets of tephra that are uniquely characteristic of Surtseyan eruptions have shapes that have been compared to cypress trees and cocks' tails and are a direct result of magma–slurry interaction [1].

The component of the ejecta that we model here is the Surtseyan bomb, a lump of magma which trails a black tail of tephra as it shoots out of an erupting tephra mass, hence appearing at the tips of the cypress tree tephra shapes illustrated in figure 1. This black tail turns white within seconds, because of cooling and condensation of superheated water vapour within it [1,4]. Examination of bombs collected from Surtseyan deposits [4] reveals that they are vesicular

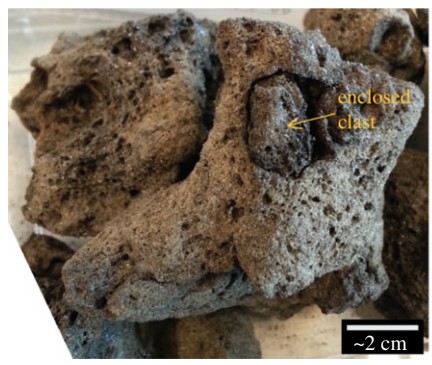
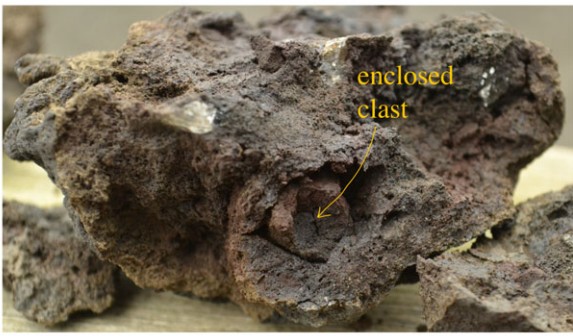

**Figure 2.** Composite clasts from Surtsey (Iceland), showing entrained smaller clasts near the outer surface or after breaking off a piece of bomb. (Online version in colour.)

(bubbly) with porosities ranging from 0.35 to 0.8, and they are hosts for entrained or engulfed smaller groups of clasts. The vesicularity is due to the usual process of water coming out of solution in the magma as a result of pressures dropping as the magma rises in the throat of the volcano. Samples of bombs are shown in figure 2, and there is a video in the repository referenced in [4] showing multiple inclusions in virtual X-ray sections through a bomb. Each enclosed clast is usually found to be loosely held in a void space in its host bomb. The first few millimetres of the material surrounding a void show evidence of compression and densification, suggestive of high historical pressures [4] and indicative of the magma having been above the glass transition temperature for some time after slurry entrainment [10]. The bombs are permeable to fluid flow, owing to their high vesicularity [11].

If a volume of liquid is flashed instantaneously to steam while constrained to that same volume, the resulting pressure [12] far exceeds the tensile strength of the surrounding vesicular magma and the bomb is then expected to fragment. However, observations indicate that many bombs survive intact, suggesting that while steam is created by heat transfer from magma, raising the pressure inside a bomb, it can also escape through the surrounding vesicular bomb, relieving the pressure increase. So there is a race between heating that creates steam to raise the pressure and steam flow that relieves the pressure inside a bomb. Resolving the winner of this race provides the motivation for the development of a model for the transient behaviour of pressure and temperature inside Surtseyan bombs while they are in free fall after ejection in order to find the conditions under which a bomb is expected to survive the development of high pressures around a flashing slurry inclusion.

Other possible fragmentation mechanisms that we do not address here include impact with the ground and vesiculation of the interior together with quenching of the outer surface.

The most relevant previous work modelling the flashing of liquid inside vesicular magma is a relatively simple model [12] that takes a cavalier approach in ignoring temperature transients, despite their importance in driving up the pressure. By ignoring the possibly very large initial temperature gradients that drive the flashing of liquid to steam, that work is likely to lead to a serious underestimate of the maximum pressures that arise. The main purpose of the present work is to properly include the driving mechanism of transient temperature gradients at the boiling front surrounding a slurry inclusion, and hence provide more reliable estimates for the maximum pressures that are developed in the inherently transient interaction between heating and steam escape.

There is no other directly comparable previous work on modelling pressure increase inside Surtseyan bombs that we are aware of. The style of modelling here is inspired by that used to explain shock tube experiments [13] in which volcanic rock samples are fragmented by pressure differences that propagate into the samples as high-pressure gas escapes from them. However, in

that modelling and experiments the thermal behaviour is adiabatic, in contrast to the transient energy conservation principles needed here.

A similar approach can be seen in recent modelling of the flashing of liquid when frying potato snacks [14]. The flashing of liquid water to steam is central to that modelling, but the development of high pressures is not of interest because of the relatively large permeabilities of the cooked snack, so that those authors are able to use a steady-state diffusion equation for vapour flow outwards from the flashing front.

As in the model presented in [12], we make no attempt here to model the development of vesicularity in rising magma or to consider viscous flow effects or changes in porosity due to compression. Instead, vesicular molten magma is treated as a competent porous rock matrix of constant porosity, owing to the high viscosity of the magma at temperatures at and below 1275 K [10,15,16]. The Deborah number

$$\mathrm{De} = \frac{t_{\mathrm{vis}}}{t_{\mathrm{elas}}}$$

may be used to separate the time scales on which viscous (flowing) and elastic (brittle) responses occur in magma under stress. Here $t_{\mathrm{vis}}$ is the time required for viscous relaxation of magma and $t_{\mathrm{elas}}$ is the time needed to deform magma as a competent solid. Brittle responses occur for $\mathrm{De} > 1$ [16], and viscous flow is more important for relieving stress if $\mathrm{De} < 0.01$. Viscous relaxation times are strongly dependent on temperature, and at 1275 K $t_{\mathrm{vis}} \sim 1$ s. Then the assumption of brittle behaviour is good for

$$\mathrm{De} \sim \frac{1}{t_{\mathrm{elas}}} > 1,$$

that is, for times $t_{\mathrm{elas}} < 1$ s. It will be seen in the numerical simulations that maximum pressure is reached in a millisecond or less, consistent with this assumption.

In the remainder of this paper, we present new measurements of porosity and permeability in intact Surtseyan bombs, and a regression relationship between them. We then build a new fully transient model from scratch by starting with physically accurate and properly coupled conservation equations in the vapour and liquid regions, with a moving boiling front between them. Once we have obtained a consistent set of coupled nonlinear partial differential equations, we non-dimensionalize, choosing appropriate scalings so that the essential transport mechanisms are captured in a reduced set of equations. A particular focus in this paper is to then explore the theoretical consequences for the maximum pressure developed in the model, if the initial temperature profile is allowed to have a step change at the flashing front. We combine numerical solutions with asymptotic arguments which suggest that the step change initial temperature case is mathematically ill-posed. An initial temperature profile that is ramped from cold to hot over a distance the size of a typical pore provides a criterion for fragmentation in terms of magma permeability or porosity. The criterion indicates that the permeabilities of the intact bombs are all high enough that the maximum pressures reached are not expected to fragment the bombs.

## 2. Fieldwork and data

Porosity and permeability were determined on small (<64 mm diameter) ejecta collected from the island of Surtsey, Iceland [17]. Deposits formed during typical Surtseyan tephra jetting were examined in the field in 2015. The deposits consisted of poorly sorted discontinuous beds, 10–20 cm in thickness. They were dominated by the fine ash (less than 2 mm) that typifies the products of phreatomagmatic eruptions [18], but also included abundant lapilli (2–64 mm) and strikingly outsized bombs (greater than 64 mm).

Bombs were universally observed to be composites, containing identifiable remnants of slurry incorporation (figure 2). The slurry inclusions are usually of a relatively large grainsize, but there is no reason to believe that the overall particle size distribution of the slurry itself should be any different from that in the preserved deposits. Our porosity and permeability data are from lapilli within this deposit, as they are the most representative of portions of vesicular magma that evaded fragmentation and they retain the textural characteristics of the magma as it erupted. Ash

particles are too small to retain useful information about pore networks, and bombs have greater potential for post-eruptive expansion that can overprint the syn-eruptive magmatic textures that are of interest here [19].

The lapilli were examined using X-ray computed microtomography (μ-cT). Each sample was scanned with a Phoenix Nanotom 180 X-ray μ-cT at l'Université d'Orléans, Orléans, France. Totals of 2000–2300 scans of each sample were collected during 360° rotation, using a tungsten filament and molybdenum target. Operating voltages were in the range 80–100 keV, with currents of 50–90 nA. Voxel edge lengths ranged from approximately 2.5 to 6.5 μm. Raw scans were reconstructed into stacks of greyscale images with an offline PC microcluster running Phoenix reconstruction software. Quantification of porosity and modelling of permeability were performed on representative elementary volumes of 500–750 px$^3$ isolated within each μ-cT scan. Multiple sub-volumes capturing the range of vesicle heterogeneity in natural samples were isolated and analysed separately, and all isolated sub-volumes were chosen to avoid any entrapped slurry or other inclusions that were not representative of the magmatic foams themselves. Vesicles were measured using the three-dimensional object counter plugin for ImageJ [20], after greyscale thresholding to isolate void space from glass and phenocrysts.

We modelled permeability using a gas flow simulation program [21]. This parallel computing program measures Darcian permeability by simulating single-phase gas flow with lattice Boltzmann simulations on the Palabos computational fluid dynamics platform. For each sub-volume, three calculations were made in three orthogonal directions. Simulations were conducted using low inlet pressures, ensuring low Reynolds number flow conditions, thus neglecting inertial contributions to total permeability [22]. The investigated samples are marginally smaller than the bombs described in this work and shown in figure 2, but have similar matrix porosities, and are considered to be unfragmented, intact bodies that are representative of Surtseyan magma at the time of eruption.

These calculations of permeability and porosity of the vesicular network within intact, unfragmented samples of Surtseyan ejecta are summarized in figure 3, together with the fitted straight line

$$\log_{10} k = 6.4\phi - 14.1. \tag{2.1}$$

This relationship between permeability and porosity is similar to that observed in other studies of vesicular basaltic magma [11], and is used later when analysing model behaviour. Even though some vesicles adjacent to slurry domains in composite bombs appear modestly compressed, we consider this to not devalue the use of the porosity–permeability relationship shown in figure 2, because the compression occurs in a narrow halo around slurry domains and because the hysteresis effect on permeability in vesicular magmas ensures that modestly compressed bubble networks will retain significant permeabilities developed during their previously uncompressed states [22].

## 3. Mathematical model

We model the boiling of the water at the surface of a single slurry inclusion, due to the surrounding relatively hot vesicular magma, and the subsequent transport of vapour out of the magma. We focus on the pressure changes in connected pores consequent on boiling, in an approach which is simpler than that in [13], positing that an excessive pore pressure will fragment the surrounding magma and making no attempt to model stress and strain in the magma itself.

We represent both slurry and magma as nested equivalent spheres of solid porous material. We place the inclusion at the centre of the magma and set this centre to be the origin, giving a frame of reference that is travelling with the bomb. The entire Surtseyan bomb is ejected from the volcanic vent and is travelling through the air in free fall for the first several seconds of its existence, so gravity effects are ignored. We neglect any effects from possible rotation of the bomb or air friction on the outside of the bomb. The slurry inclusion is modelled as a porous medium filled with liquid water. The slurry temperature is assumed to be initially near boiling point at

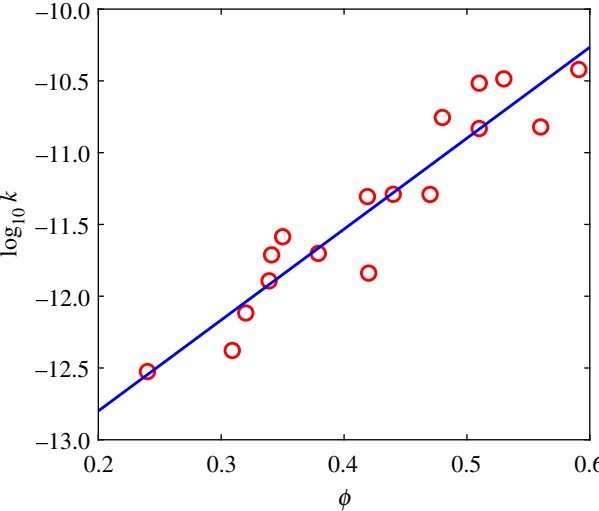

**Figure 3.** The log of permeability plotted against porosity, as measured in samples of Surtseyan ejecta, together with a best fit straight line. (Online version in colour.)

atmospheric pressure. Our simulations indicate that initial slurry temperature has little effect on pressure development compared with the magma temperature. The enveloping hot magma is treated as a porous medium with its connected porosity containing only steam. A two-phase region will develop at the interface between liquid and steam phases. We assume, supported later by the large size of the Stefan number, that the two-phase region is relatively thin. The entire bomb is assumed to start at atmospheric pressure.

The pointwise conservation of enthalpy equation for a moving fluid (which here may be the liquid or vapour phase of water) in the absence of sources or sinks takes the form [23]

$$\frac{\partial}{\partial t}(\varrho h) + \nabla \cdot (\varrho h \mathbf{v}) = -\nabla \cdot \mathbf{q} + \frac{\mathrm{D}p}{\mathrm{D}t} + \tau : \nabla \mathbf{v}, \tag{3.1}$$

where $\varrho$ is the fluid density, $h$ is the specific enthalpy, $p$ is the fluid pressure, $\mathbf{v}$ is the local fluid velocity vector, $\mathbf{q}$ is the heat flux and $\mathrm{D}/\mathrm{D}t = \partial/\partial t + \mathbf{v} \cdot \nabla$ is the total derivative operator. The symbol $\tau$ is the deviatoric stress tensor in the viscous dissipation term.

As noted in [24] and in appendix D of [25], the pointwise conservation of enthalpy equation can be written in terms of pressure and temperature $T$ as

$$\varrho c_p \frac{\mathrm{D}T}{\mathrm{D}t} - \beta T \frac{\mathrm{D}p}{\mathrm{D}t} = \nabla \cdot (\kappa_T \nabla T) + \tau : \nabla \mathbf{v}, \tag{3.2}$$

where the specific heat at constant pressure is

$$c_p = T \left( \frac{\partial S}{\partial T} \right)_p,$$

the specific enthalpy is $h = U + p/\varrho$, $S$ is the specific entropy and $U$ is the specific internal energy. We have also used $\mathrm{d}h = T\,\mathrm{d}S + \mathrm{d}p/\varrho$ and $T\,\mathrm{d}S = c_p\,\mathrm{d}T - (\beta T/\varrho)\,\mathrm{d}p$ to obtain equation (3.2). The coefficient of isothermal compressibility is

$$\beta = \varrho \left( \frac{\partial(1/\varrho)}{\partial p} \right)_T.$$

We have also used Fourier's law for heat conduction $\mathbf{q} = -\kappa_T \nabla T$, where $\kappa_T$ is the thermal conductivity of the fluid, and the mass conservation equation for fluid,

$$\frac{\partial \varrho}{\partial t} + \nabla \cdot (\varrho \mathbf{v}) = 0. \tag{3.3}$$

When considering the rock component of the porous medium that we take to comprise both the slurry inclusion and the surrounding hot magma, we ignore small displacements, velocities and compressibility, and we consider it to be a competent solid material. We assume our representative elementary volume is small enough that there are no appreciable local temperature differences between rock and steam, so that, after averaging as in [23,25] over a representative elementary volume of porosity $\phi$, we have conservation of energy in the rock component as

$$\varrho_m c_{pm}(1 - \phi)\frac{\partial T}{\partial t} = (1 - \phi)\nabla \cdot (\kappa_m \nabla T),\qquad(3.4)$$

where $\varrho_m$ is the rock density in the absence of any porosity, $c_{pm}$ is the specific heat capacity of the rock and $(1 - \phi)\kappa_m$ is its effective thermal conductivity.

We now consider separately the liquid-filled slurry region $r < s(t)$ inside the flashing front and the vapour-filled hot magma region $R_2 > r > s(t)$ outside the flashing front. The only distinction we make between slurry and hot magma is that cool liquid occupies the porosity in the slurry region and hot vapour occupies the porosity in the magma region. For computational simplicity here, but also consistent with our observations in §2 that inclusions have similar properties to surrounding rock, we assume the same (constant) porosity $\phi$ in both regions.

## (a) Slurry region

Conservation of mass for liquid water in the slurry region $0 < r < s(t)$ is given by

$$\phi\frac{\partial \varrho_l}{\partial t} + \nabla \cdot (\varrho_l \mathbf{u}_l) = 0,\qquad(3.5)$$

and the momentum conservation equation for liquid is given by Darcy's law, as liquid velocities arising from the compressibility of liquid water are found to be small,

$$\mathbf{u}_l = -\frac{k}{\mu_l}\nabla p,\qquad(3.6)$$

where $p$ is the pressure in the fluid, and we do not model pressure in the rock matrix. An equation of state for liquid density is provided by the expression

$$\varrho_l = \varrho_{0l} + \beta_\ell(p - p_a) - \alpha(T - T_0^w),\qquad(3.7)$$

which fits within a 10% accuracy the specific volume data present in [26] for pressures in the range 1–100 bars and temperatures between saturation and 573 K. The terms $\varrho_{0l}$, $p_a$ and $T_0^w$ are reference values of liquid density, pressure and temperature, listed with $\alpha$, $\beta_\ell$ in table 1. Averaging the pointwise energy equation (3.2) over a representative elementary volume as in [23,25], the energy equations for liquid water and solid rock combine to give

$$\varrho'c'\frac{\partial T}{\partial t} + \varrho_l c_{\mathrm{pl}}\mathbf{u}_l \cdot \nabla T - \phi\beta T\frac{\partial p}{\partial t} - \beta T\mathbf{u}_l \cdot \nabla p = \kappa_{\mathrm{el}}\nabla^2 T + \overline{\tau : \nabla\mathbf{u}_l},\qquad(3.8)$$

where the temperature $T$ is assumed to be the same in fluid and adjacent rock, $\varrho_l$ is the density of liquid water, $\mathbf{u}_l = \phi\mathbf{v}_l$ is the Darcy velocity of the liquid water, $\kappa_{\mathrm{el}} = (1 - \phi)\kappa_m + \phi\kappa_l$ is the average thermal conductivity of the liquid-filled region, $c_{\mathrm{pl}}$ is the specific heat of liquid water and $\kappa_l$ is the thermal conductivity of liquid water. Typical values for these parameters are given in table 1. The symbol $\tau$ is the deviatoric stress tensor in the averaged viscous dissipation term, accounting for heat generated in liquid water as a result of visous shear flow within pores.

## (b) Magma region

In the region $R_2 > r > s(t)$, there is only steam in the pores, mostly above the critical temperature. Initially, this steam would be magmatic vapour from the vesiculation process that created the pores, which we are not modelling. It will be displaced by steam generated by flashing the liquid

**Table 1.** Physical constants.

| constant | name | typical value | units |
|---|---|---|---|
| $c_{pl}$ | specific heat of liquid water | 4200 | $J\,kg^{-1}\,K^{-1}$ |
| $c_m$ | specific heat of magma | 840 | $J\,kg^{-1}\,K^{-1}$ |
| $c_{ps}$ | specific heat of steam | 2000 | $J\,kg^{-1}\,K^{-1}$ |
| $h_{sl}$ | specific heat of vaporization | $2.3 \times 10^6$ | $J\,kg^{-1}$ |
| $k$ | permeability | $10^{-12}$ | $m^2$ |
| $\kappa_e$ | thermal conductivity [27] | 2 | $W\,m^{-1}\,K^{-1}$ |
| $\kappa_{el}$ | thermal conductivity [27] | 3 | $W\,m^{-1}\,K^{-1}$ |
| $M$ | molar mass of water | $18 \times 10^{-3}$ | $kg\,mol^{-1}$ |
| $p_a$ | atmospheric pressure | $10^5$ | $Pa$ |
| $R$ | universal gas constant | 8.314 | $J\,K^{-1}\,mol^{-1}$ |
| $R_1$ | initial inclusion radius | 0.01 | m |
| $R_2$ | magma (bomb) radius | 0.1 | m |
| $T_i$ | initial inclusion temperature | 373 | K |
| $T_m$ | initial magma temperature | 1275 | K |
| $T_c$ | critical temperature of water | 647 | K |
| $T_0^w$ | reference temp, $T_i$ | | |
| $T_0^e$ | reference temp, $T_i$ | | |
| $\alpha$ | thermal expansion coefficient for liquid water | 0.5 | $kg\,m^{-3}\,K^{-1}$ |
| $\beta_\ell$ | isothermal compressibility of liquid water | $4.6 \times 10^{-10}$ | $kg\,m^{-3}\,Pa^{-1}$ |
| $D_m$ | thermal diffusivity $\kappa_e/(\varrho c)$ in magma | $1.4 \times 10^{-6}$ | $m^2\,s^{-1}$ |
| $\mu_v$ | dynamic viscosity of water vapour | $3 \times 10^{-5}$ | $Pa\,s$ |
| $\phi$ | porosity | 0.3 | |
| $\varrho_m$ | density of basalt | 2750 | $kg\,m^{-3}$ |
| $\varrho c$ | $(1-\phi)\varrho_m c_m + \phi\varrho_s c_{ps}$ | $1.6 \times 10^6$ | $J\,m^{-3}\,K^{-1}$ |
| $\varrho' c'$ | $(1-\phi)\varrho_m c_m + \phi\varrho_l c_{pl}$ | $3 \times 10^6$ | $J\,m^{-3}\,K^{-1}$ |
| $\varrho_{0l}$ | reference liquid density | 1000 | $kg\,m^{-3}$ |

in the slurry, the process we are modelling here. Mass conservation of steam is given by

$$\phi \frac{\partial \varrho_s}{\partial t} + \nabla \cdot (\varrho_s \mathbf{u}) = 0. \tag{3.9}$$

The momentum conservation equation for fluid flow is typically given by Darcy's law for laminar flow in a porous medium. As it is possible that steam flow might be turbulent, we use the Forchheimer equation for steam pressure $p$, which combines Darcy flow with the turbulent Ergun equation [28],

$$\nabla p = -\frac{\mu_v}{k}\mathbf{u} - \frac{\varrho_s c_F}{\sqrt{k}}\mathbf{u}|\mathbf{u}|, \tag{3.10}$$

where $c_F$ is an order 1 coefficient and $\varrho_s$ is the density of steam.

For an ideal gas, $\beta T = 1$. Averaging the pointwise energy equation (3.2) for steam and the rock energy equation (3.4) over a representative elementary volume of the porous magma gives the

averaged energy equation for steam and rock,

$$\varrho c \frac{\partial T}{\partial t} + \varrho_s c_{\mathrm{ps}} \mathbf{u} \cdot \nabla T - \phi \frac{\partial p}{\partial t} - \mathbf{u} \cdot \nabla p = \kappa_e \nabla^2 T + \overline{\tau : \nabla \mathbf{u}}, \tag{3.11}$$

where the temperature $T$ in steam is assumed to be equal to that in adjacent rock, $\mathbf{u} = \phi \mathbf{v}$ is the Darcy velocity of steam, $\kappa_e = (1 - \phi)\kappa_m + \phi\kappa_s$ is the effective thermal conductivity of magma with steam in the pores and $\kappa_s$ is the thermal conductivity of steam.

The last (drag) term is the viscous dissipation in the steam flow, averaged over a representative elementary volume. This is the heat generated by viscosity in shearing flow for steam as it moves through porous magma. This term is often neglected [25, E.4]. We estimate its size here using eqn 10.7.24 in [29] (see also [30]) and Darcy's law,

$$\overline{\tau : \nabla \mathbf{u}} = -\frac{\mu}{k}\mathbf{u}^2 = -\frac{k}{\mu}(\nabla p)^2.$$

The pressure gradient is estimated using a maximum pressure difference that equals a tensile strength of 1 MPa for magma over a length scale of 0.1 m to give a magnitude for averaged viscous dissipation of the order of $10^6$. This is a factor of $10^3$ smaller than the sensible heating rate $\varrho c$ $(\partial T/\partial t) \sim 10^9$, using a time scale of 1 s, and a factor of $10^2$ smaller (on a length scale of one pore) than the diffusion term, $\kappa_e \nabla^2 T \sim 10^8$. So we neglect viscous dissipative heating due to steam drag in the magma region, compared with heating due to diffusion from nearby hot magma.

The variation of $\varrho c$ due solely to changes in steam temperature of $1200°C$ and changes in steam pressure of 20 bars (typical tensile strength of volcanic rock) is less than 0.5%. It is dominated by the thermal capacity of the magma, which typically varies from values near $800\,\mathrm{J\,kg}^{-1}\,\mathrm{K}^{-1}$ at 400 K, rising rapidly to values near $1300\,\mathrm{J\,kg}^{-1}\,\mathrm{K}^{-1}$ at glass temperatures near 900–1000 K [31]. We have assumed a constant value for $\varrho c$, neglecting variations of $\pm 30\%$.

The ideal gas law is used in the vapour region, so that, for $r > s(t)$,

$$\varrho_s = \frac{pM}{RT}. \tag{3.12}$$

## (c) Flashing front

We assume that boiling occurs in a thin moving region located at $r = s(t)$ that separates the liquid and vapour regions. We acknowledge the spherical symmetry of our model by taking all variables to depend only on $r$ and $t$. Then vapour and liquid velocities have only a radial component $u = \phi v$, $u_l = \phi v_l$, respectively. We write the energy equations in enthalpy form, and integrate them and the mass conservation equations with respect to volume across the moving flashing front, to obtain

$$\phi \varrho_s h_{\mathrm{sl}}(v - \dot{s}) = \phi \varrho_l h_{\mathrm{sl}}(v_l - \dot{s}) = [\kappa \nabla T]_-^+ + \phi(v - v_l)p, \tag{3.13}$$

where $h_{\mathrm{sl}} = h_s - h_l$ is the specific heat of vaporization and

$$[\kappa \nabla T]_-^+ = \kappa_e \nabla T(0^+) - \kappa_{\mathrm{el}} \nabla T(0^-).$$

At the flashing front, pressure and temperature are related by the Clausius–Clapeyron equation,

$$p = p_0^e \mathrm{e}^{[Mh_{\mathrm{sl}}/(RT_0^e)][(T - T_0^e)/T]}, \tag{3.14}$$

where $T_0^e$ and $p_0^e$ are the reference temperature and pressure values for the liquid and vapour phases of water at equilibrium.

These equations (3.5)–(3.14) form our dimensional model equations. Boundary conditions are

$$T(R_2) = 300\,\mathrm{K}, \quad p(R_2) = p_a, \quad \frac{\partial T}{\partial r} = \frac{\partial p}{\partial r} = 0 \text{ at } r = 0.$$

Temperature and pressure are assumed to be continuous across the flashing front. Initial conditions are that the temperature of the magma is $T_m$ and the temperature of the inclusion is at boiling point for atmospheric pressure, $T_i$. Initial pressures are taken to be $p_a$ everywhere.

## (d) Fragmentation criterion

We use the criterion for fragmentation that steam pressure exceeds the critical value

$$p_c = (1 - \phi)\sigma_Y, \tag{3.15}$$

where $\sigma_Y = 2\,\text{MPa}$ is a typical value for tensile strength of magmatic rock. This criterion arises out of the use of a soil mechanics approach in shock tube fragmentation research and is presented in [13], which is also based in part on Biot's work on wave propagation in fluid-filled porous media [32]. The factor $(1 - \phi)$ accounts for the fraction of the magma that is load-bearing. A similar criterion is recommended in other studies of shock tube fragmentation of volcanic rock, including [33, eqn 11] and [34].

# 4. Model reduction

We rescale (see also table 1) using

$$r = \tilde{r}R_2, \quad s = \tilde{s}R_2, \quad T = \tilde{T}T_m, \quad p = \tilde{p}p_a, \quad t = \tilde{t}t_0,$$
$$v_l = v_{0l}\tilde{v}_l, \quad v = v_{0s}\tilde{v}, \quad \varrho_l = \varrho_{0l}\tilde{\varrho}_l, \quad \varrho_s = \varrho_{0s}\tilde{\varrho}_s.$$

Noting that flashing of liquid to vapour is the driving force for pressure change inside the bomb, the time scale $t_0$ is chosen to be the time required to flash all of the liquid in the slurry inclusion to steam. A balance between the energy required to do this and the heat provided by conduction across the slurry surface at $r = R_1$ gives the estimate $t_0 = \phi\varrho_{0l}h_{sl}R_1^2/(3\kappa_e(T_m - T_i)) \approx 17$.

We use the ideal gas law to provide $\varrho_{0s} = p_a M/(RT_m)$, and since the source of steam is the flashing front we balance $\dot{s}\varrho_l$ with $v_s\varrho_s$, giving

$$v_{0s} = \frac{R_2\varrho_{0l}}{t_0\varrho_{0s}}. \tag{4.1}$$

In the following equations, we drop the tilde on dimensionless variables for simplicity of notation. The critical pressure for fragmentation is rescaled to give the non-dimensional value

$$p_c = 20(1 - \phi). \tag{4.2}$$

## (a) In the liquid region

In the slurry, the equation of state (3.7) non-dimensionalizes to

$$\varrho_l = 1 + \delta_1(p - 1) - \lambda_1(T - T_0). \tag{4.3}$$

Owing to the symmetry of the problem the only mechanisms that can cause liquid to move in the slurry are liquid density changes. The relatively small size of $\delta_1$ (table 2) means that thermal expansion is the main factor. We over-estimate the velocity scale $v_{0l}$ by calculating the change in dimensional radius due to thermal expansion as the slurry is heated to near-critical temperature $T_{\text{crit}}$, and the time to heat the ball to this temperature by conduction. Ignoring losses of liquid from the slurry, the slurry mass is the volume multiplied by the density, so that if the slurry starts at radius $R_1$, density $\varrho_{0l}$ and temperature $T_0^w$, and expands to a new radius $R_1 + \Delta R_1$, then $(R_1 + \Delta R_1)^3[\varrho_{0l} - \alpha(T_{\text{crit}} - T_0^w)] = R_1^3\varrho_{0l}$. For small $\Delta R_1$ this gives $\Delta R_1 \approx R_1\alpha(T_{\text{crit}} - T_0^w)(3(\varrho_{0l} - \alpha(T_{\text{crit}} - T_0^w)))$. Estimating the time $t_s$ required to heat the slurry by conduction gives $t_s \approx R_1^2\varrho c/\kappa_{\text{el}}$, so that the liquid velocity scale is approximated by

$$v_{0l} \approx \frac{\Delta R_1}{t_s} = \frac{\alpha\kappa_{\text{el}}(T_{\text{crit}} - T_0^w)}{3(\varrho_{0l} - \alpha(T_{\text{crit}} - T_0^w))R_1\varrho c} \approx 10^{-5}\,\text{m s}^{-1}.$$

The conservation of mass equation in the slurry becomes, after rescaling and dropping the tildes,

$$\phi\frac{\partial\varrho_l}{\partial t} + v_{\text{dl}}\frac{\partial}{\partial r}(\varrho_l u_l) = 0. \tag{4.4}$$

**Table 2.** Parameters.

| parameter | definition | typical value | units |
|---|---|---|---|
| $\delta_1$ | $\beta_\ell p_a/\varrho_{0l}$ | $4.6 \times 10^{-8}$ | |
| $\delta_2$ | $t_0 v_{0s}\phi\varrho_{0s}c_{ps}/(R_2\varrho c)$ | 0.4 | |
| $\delta_3$ | $\phi p_a/(\varrho c T_m)$ | $1.5 \times 10^{-5}$ | |
| $\delta_4$ | $\phi v_{0s}p_a t_0/(R_2\varrho c T_m)$ | 0.09 | |
| $\delta_5$ | $\kappa_e t_0/(\varrho c R_2^2)$ | $2 \times 10^{-3}$ | |
| $\epsilon_\varrho$ | $\varrho_{0s}/\varrho_{0l}$ | $1.7 \times 10^{-4}$ | |
| $\epsilon_1$ | $v_{0l}/v_{0s}$ | $2 \times 10^{-7}$ | |
| $\epsilon_2$ | $\phi\beta_\ell p_a/(\varrho' c')$ | $5 \times 10^{-12}$ | |
| $\epsilon_3$ | $\kappa_{el}t_0/(\varrho' c' R_2^2)$ | $10^{-3}$ | |
| $\epsilon_4$ | $k p_a/(\mu_v R_2 \phi v_{0s})$ | $2 \times 10^{-3}$ | |
| $\epsilon_5$ | $k p_a t_0/(\phi\mu_v R_2^2)$ | 14 | |
| $\epsilon_6$ | $\sqrt{k}\varrho_{0s}c_F\phi v_{0s}/\mu_v$ | 0.08 | |
| $\lambda_1$ | $\alpha T_m/\varrho_{0l}$ | 0.6 | |
| $\lambda_2$ | $p_a/(\varrho_{0s}h_{sl})$ | 0.26 | |
| $\varrho_{0s}$ | $p_a M/(R T_m)$ | 0.17 | kg m$^{-3}$ |
| $\varrho_{0l}$ | reference value of $\varrho_l$ | 1000 | kg m$^{-3}$ |
| $H$ | $M h_{sl}/(R T_0^e)$ | 13 | |
| St | $h_{sl}\phi\varrho_{0s}R_2 v_{0s}/(T_m\kappa_e)$ | 212 | |
| $t_0$ | $\phi\varrho_{0l}h_{sl}R_1^2/(3\kappa_e(T_m - T_i))$ | 13 | s |
| $T_0$ | $T_i/T_m$ | 0.29 | |
| $v_{0s}$ | $R_2\varrho_{0l}/(t_0\varrho_{0s})$ | 46 | m s$^{-1}$ |
| $v_{0l}$ | $\alpha\kappa_{el}(T_c - T_0^w)/(3R_1\varrho c(\varrho_{0l} - \alpha(T_c - T_0^w)))$ | $10^{-5}$ | m s$^{-1}$ |
| $v_{dl}$ | $v_{0l}t_0/R_2$ | $10^{-3}$ | |

Because $v_{dl} \sim 10^{-3} \ll 1$ and $u_l = 0$ at the origin, equation (4.4) implies that, to leading order in $v_{dl}$, $\varrho_l \sim 1$ and $u_l = 0$. Then pressures in the slurry do not vary appreciably with radius and can be taken to be the same as the time-varying pressure value at the flash front. It follows also that $\varrho' c'$, which depends mainly on $\varrho_l$, can now be treated as a constant.

The conservation of energy equation (3.8) in the slurry rescales to give, after dropping the tildes and neglecting terms involving liquid velocity,

$$\frac{\partial T}{\partial t} - \epsilon_2 T\frac{\partial p}{\partial t} = \frac{\epsilon_3}{r^2}\frac{\partial}{\partial r}\left(r^2\frac{\partial T}{\partial r}\right), \tag{4.5}$$

where the parameters $\epsilon_2$, $\epsilon_3$ are given in table 2. The parameter $\epsilon_2 \sim 10^{-11}$ is eight orders of magnitude smaller than $\epsilon_3 \sim 10^{-3}$, indicating that this pressure–work term may be neglected. However, we retain for now the term with the small parameter $\epsilon_3$, anticipating that the net temperature gradient in the thermal boundary layer at the flashing front is what drives changes in pressure. Hence in the slurry, we reduce to the energy equation

$$\frac{\partial T}{\partial t} = \frac{\epsilon_3}{r^2}\frac{\partial}{\partial r}\left(r^2\frac{\partial T}{\partial r}\right), \quad r < s(t). \tag{4.6}$$

## (b) In the vapour region

In the hot magma vapour transport region $r > s(t)$, Forchheimer's equation (3.10) non-dimensionalizes to $\epsilon_4 \nabla p = -v - \epsilon_6 \varrho_s v |v|$. Since $\epsilon_6 \approx 0.008 \ll 1$ (table 2), the Ergun term is not required,[1] and Forchheimer's equation reduces to Darcy's law,

$$v = -\epsilon_4 \nabla p. \tag{4.7}$$

The mass equation (3.9) becomes, after non-dimensionalising and substituting for $v$,

$$\frac{\partial \varrho_s}{\partial t} = \epsilon_5 \nabla \cdot (\varrho_s \nabla p). \tag{4.8}$$

The energy equation (3.11) takes the non-dimensional form

$$\frac{\partial T}{\partial t} + \delta_2 \varrho_s v \frac{\partial T}{\partial r} - \delta_3 \frac{\partial p}{\partial t} - \delta_4 v \frac{\partial p}{\partial r} = \frac{\delta_5}{r^2} \frac{\partial}{\partial r} \left( r^2 \frac{\partial T}{\partial r} \right), \tag{4.9}$$

where the meanings and values of the parameters are given in table 2.

Using Darcy's law (4.7) to replace $v$ gives

$$\frac{\partial T}{\partial t} - \epsilon_4 \delta_2 \varrho_s \frac{\partial p}{\partial r} \frac{\partial T}{\partial r} - \delta_3 \frac{\partial p}{\partial t} + \delta_4 \epsilon_4 \left( \frac{\partial p}{\partial r} \right)^2 = \frac{\delta_5}{r^2} \frac{\partial}{\partial r} \left( r^2 \frac{\partial T}{\partial r} \right). \tag{4.10}$$

The most significant terms in the above equation give a diffusion equation for temperature,

$$\frac{\partial T}{\partial t} = \frac{\delta_5}{r^2} \frac{\partial}{\partial r} \left( r^2 \frac{\partial T}{\partial r} \right), \quad r > s(t). \tag{4.11}$$

We retain the diffusion term containing the small parameter $\delta_5 \sim 1.7 \times 10^{-3}$, as we need the temperature gradients in the thermal boundary layers at the flashing front to calculate the speed of the flashing front. We have neglected the heat advection term involving $\epsilon_4 \delta_2 \sim 7.4 \times 10^{-4}$, which is nearly half the size of the diffusion term. We have also neglected the pressure–work terms involving $\delta_3$ and $\delta_4 \epsilon_4$, which are $10^{-2}$ times $\delta_5$. We have checked *a posteriori* that they remain relatively small despite pressure increases of order 10 simulated later in this paper.

The ideal gas law takes the non-dimensional form

$$p = \varrho_s T. \tag{4.12}$$

## (c) At the flashing front

Conservation of mass and energy across the flashing front $r = s(t)$ as expressed in equations (3.13) give

$$\dot{s}(\varrho_l - \epsilon_\varrho \varrho_s) = -\varrho_s v + v_{\mathrm{dl}} \varrho_l v_l \tag{4.13}$$

and

$$\varrho_s(v - \epsilon_\varrho \dot{s}) = \frac{1}{\mathrm{St}} [\nabla T]_-^+ + \lambda_2 (v - \epsilon_1 v_l) p, \tag{4.14}$$

where constants are defined in table 2. We have neglected the small relative difference between $\kappa_e = 2$ and $\kappa_{\mathrm{el}} = 3$ to obtain equation (4.14). The Stefan number $\mathrm{St} \sim 200$ is relatively large, consistent with the assumption of a narrow two-phase region at the flashing front.

---

[1] Note that the pore Reynolds number $\mathrm{Re}_p = \rho_f \sqrt{k\phi} |u| / \mu_v$, with simulations giving $v \approx 10\,\mathrm{m\,s^{-1}}$, leading to $\mathrm{Re}_p \approx 0.01$. Hence the flow is laminar.

Noting the small size of $\epsilon_\varrho \sim 10^{-4}$ and $\epsilon_1 \sim 10^{-7}$, we drop the advective mass and heat transport terms in equations (4.13) and (4.14) and use Darcy's law (4.7) to obtain

$$\dot{s} = \epsilon_4 \varrho_s \frac{\partial p}{\partial r} = -\frac{1}{\mathrm{St}} \left[ \frac{\partial T}{\partial r} \right]_-^+ + \lambda_2 \epsilon_4 p \frac{\partial p}{\partial r}.$$

The parameter combination $\lambda_2 \epsilon_4 \sim 5 \times 10^{-4}$ is 10 times smaller than $1/\mathrm{St} \sim 5 \times 10^{-3}$, so we discard the pressure–work term involving $\lambda_2 \epsilon_4$ to give the reduced jump conditions

$$\dot{s} = \epsilon_4 \varrho_s \frac{\partial p}{\partial r} = -\frac{1}{\mathrm{St}} \left[ \frac{\partial T}{\partial r} \right]_-^+. \tag{4.15}$$

The Clausius–Clapeyron condition becomes, in dimensionless terms, if we choose the reference values to be $p_0^e = p_a$ and $T_0^e = T_i = 373\,\mathrm{K}$,

$$p = \exp \left[ H \left( \frac{T - T_0}{T} \right) \right]. \tag{4.16}$$

This condition, giving the pressure dependence of the vaporization temperature of water, is only valid for the two-phase conditions that apply at the flashing front $r = s(t)$, and require that $T \in [T_0, 0.5]$ approximately.

Our reduced system then consists of equation (4.6) for temperature diffusion in the slurry, where pressures and densities are spatially constant, equation (4.8) for nonlinear pressure diffusion (coupled with temperature) in the surrounding vapour region, equation (4.11) for temperature diffusion in the vapour region, the ideal gas law (4.12) relating density, pressure and temperature in the vapour region, with boundary conditions (4.15) and (4.16) at the moving flashing front $r = s(t)$ providing the temperature, pressure and speed $\dot{s}$ there, and boundary conditions at the origin and at the surface of the bomb

$$p = 1, \ r = 1; \quad \frac{\partial T}{\partial r} = 0, \ r = 0; \quad T = T_0, \ r = 1.$$

The temperature gradients at the flashing front provide a vapour flux boundary condition, dominated by the gradient on the hot magma side that drives pressure up at the flashing front and forces vapour outwards into the bomb. Typical initial conditions would be

$$T = T_0, \ r < s(0); \quad T = 1, \ r > s(0); \quad p = 1; \quad s(0) = \frac{R_1}{R_2},$$

with a step change in temperature at the flashing front.

## (d) Thermal boundary layers

In the magma and in the slurry, thermal diffusivities are approximately 0.002 in value, which is small compared with the nonlinear diffusivity of pressure given by $\epsilon_5 \, p \sim 70$. So we note that temperature changes are expected to propagate more slowly than pressure changes, and we seek approximations to the temperatures in the thermal boundary layers near the flashing front, which drive steam production via equation (4.15), which specifies the flux at the moving flash boundary for equation (4.8).

In the magma, we consider an inner region described by a radial coordinate $\sigma$ given by $r = \epsilon + \sqrt{\delta_5}\sigma$, which is close to the flashing front that starts at $\epsilon = R_2/R_1$, at times that are early enough to ignore movement of the flashing front. Then the temperature equation in the magma in this inner region becomes

$$T_t = \frac{1}{(\epsilon + \sqrt{\delta_5}\sigma)^2} \frac{\partial}{\partial \sigma} \left[ (\epsilon + \sqrt{\delta_5}\sigma)^2 \frac{\partial T}{\partial \sigma} \right].$$

Considering the limit $\delta_5 \to 0$, and taking the inner solution valid for $\sqrt{\delta_5}\sigma \ll \epsilon$, we have the boundary layer equation (with subscripts $t$ and $\sigma$ indicating partial derivatives)

$$T_t = T_{\sigma\sigma}, \tag{4.17}$$

with boundary conditions $T(0, t) = T_f$ and $T \to 1$ as $\sigma \to \infty$. The temperature at the flashing front $T_f \sim 0.4$ varies with time, but only by about 5%, being limited by the critical temperature. The outer solution is $T = 1$ in the rest of the magma, away from the boundary layer. We take advantage of the relatively small variation in $T_f$, by taking it to be a constant value.

The boundary layer equation (4.17) then admits a similarity solution. Considering the similarity variable $\eta = \sigma^2/t$, the partial differential equation (4.17) becomes an ordinary differential equation $4T_{\eta\eta} + (2/\eta)T_\eta + T_\eta = 0$. This is first order in $T_\eta \equiv dT/d\eta$, and may be integrated twice to obtain the solution

$$T = (1 - T_f)\, \mathrm{erf}\left(\frac{\sigma}{2\sqrt{t}}\right) + T_f, \tag{4.18}$$

where $\mathrm{erf}(x) = (2/\sqrt{\pi}) \int_0^x e^{-u^2} du$, and the constants have been chosen to match the flash temperature $T_f$ when $\sigma = 0$ and the outer solution $T = 1$ as $t \to 0$. This inner solution provides the value of the temperature gradient at the flashing front, on the magma side, as

$$\frac{\partial T}{\partial r} = \frac{1 - T_f}{\sqrt{\pi \delta_5 t}} \approx \frac{0.6}{\sqrt{\pi \delta_5 t}}. \tag{4.19}$$

The same approach with $\sigma_2 = \frac{\epsilon - r}{\sqrt{\epsilon_3}}$ provides the temperature in the boundary layer in the slurry,

$$T = T_f + (T_0 - T_f)\, \mathrm{erf}\left(\frac{\sigma_2}{2\sqrt{t}}\right). \tag{4.20}$$

Then the temperature gradient on the slurry side of the flashing front is estimated as

$$\frac{\partial T}{\partial r} = \frac{T_f - T_0}{\sqrt{\pi \epsilon_3 t}} \approx \frac{0.04}{\sqrt{\pi \epsilon_3 t}}. \tag{4.21}$$

Noting that $\epsilon_3 \approx \delta_5$, we see that the contribution of the temperature gradient in the slurry is about one-fifteenth that of the temperature gradient in the magma, owing to the smaller temperature differences between the flash temperature and the slurry temperature, and to the similar thermal diffusivities.

# 5. Numerical solutions

Our reduced system is solved using the method of lines, and coded in Matlab. The moving boundary at the flashing front is fixed in place using Landau transformations, which lead to advective terms depending on the speed of the boundary in the partial differential equations describing the reduced model. The thermal boundary layers are resolved by transforming the spatial variable adjacent to the flashing front, which is now fixed in place, in both the slurry and hot magma regions, so as to obtain greater resolution at the flashing front. Spatial derivatives are then replaced by equispaced differences in the new spatial variable. The resulting system of coupled ordinary differential equations is stiff. We found consistent results using 1200 spatial mesh points and the stiff solver ode15s in Matlab.

Typical results are plotted in figure 4, from simulations run using the parameter values listed in table 1. Pressures rise rapidly to a global maximum at the flashing front, diffusing out into the hot magma region, then slowly decaying as the liquid in the slurry boils away to nothing.

The smallest mesh size used in these simulations corresponds to 10 μm adjacent to the flashing front, a typical value for pore size in Surtseyan bombs and pumice. Initial temperatures ramp from boiling point in the slurry to magma temperatures over one pore size. Figure 4f shows the maximum pressure reached at the flashing front versus permeability. The initial temperature profile used for this plot matches the similarity solution, ramping from flash temperature to close to magma values over a distance of 10 μm, with a smallest computational mesh size corresponding to 1 μm.

In the time taken for pressures to reach their maximum value, the flashing front has moved less than $10^{-4}$ times its original position. Nevertheless, turning off the advective terms in

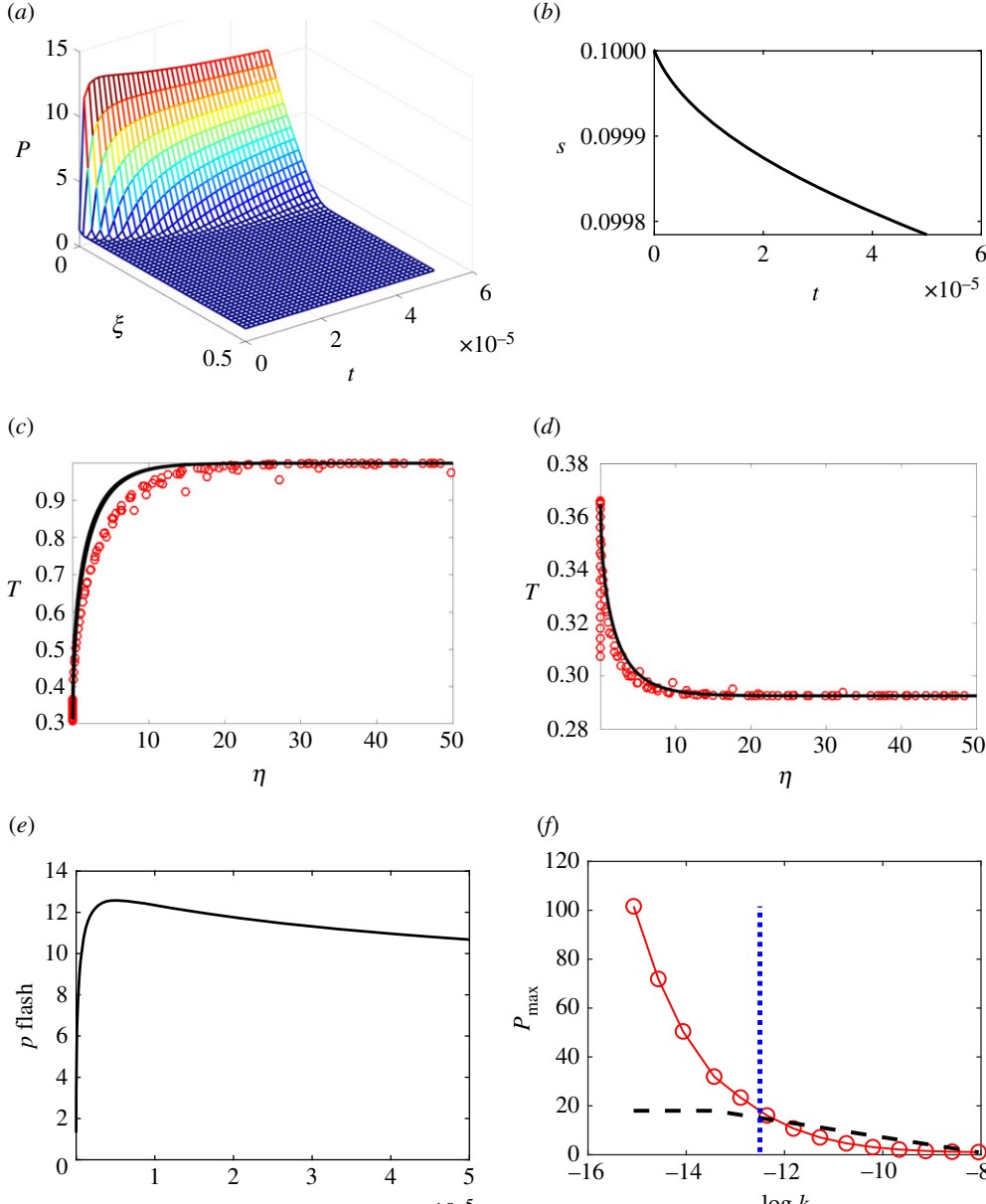

**Figure 4.** Simulation of flashing to steam inside a Surtseyan bomb. All variables are non-dimensional. Physical constant values are as listed in table 1, except in (*f*), where permeability values are taken to depend on porosity, as does the fit (2.1) to measured values for Surtseyan bombs. Eleven porosity values were chosen evenly spaced in the range [0.1, 0.95], providing the indicated values of permeability *k* (units m$^2$). The subhorizontal dashed line in (*f*) is the critical pressure (4.2), above which fragmentation occurs. The vertical dashed line indicates the smallest permeability measured in our data from intact bombs. (Online version in colour.)

the conservation equations solved numerically gives a much closer match than that shown in figure 4 to the similarity solutions for temperatures in slurry and in magma, which were derived by ignoring movement of the flashing front. The multiple temperatures plotted at $\eta = 0$ in figure 4, especially noticeable in the slurry temperature plot, correspond to the changing boiling temperature at the flashing front at early times, owing to pressure changes there.

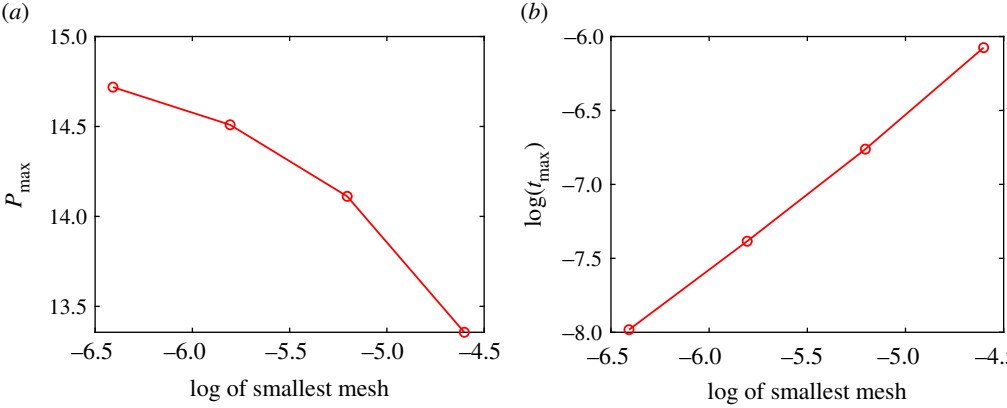

**Figure 5.** Maximum dimensionless pressure reached (*a*) and the log of the dimensionless time taken to reach it (*b*) versus the log of the dimensionless size of the smallest mesh used in computer simulations of the reduced model. Logs are to the base 10. Initial temperatures are a step function, effectively a ramp over the smallest mesh size. Parameter values used are as listed in table 1. (Online version in colour.)

Pressure rises rapidly at the flashing front because of the influx of steam there, driven mainly by the temperature gradient on the hot magma side. The initial temperature profile is at present theoretically a step function, with a gradient that is in theory infinite. In the computer code, this step change is approximated by a ramp with a steepness that depends on the smallest mesh size. The results plotted in figure 5 show how the maximum pressure at the flashing front and the time taken to reach it vary with the steepness of the initial temperature profile. The largest of these sizes used in figure 5 corresponds to a dimensional value of a few micrometres. Ten micrometres would be a typical pore size, and might be taken to be the smallest mesh size in terms of the present application. It is clear that the maximum pressure increases as the ramp tends towards a step, and that the time taken decreases roughly proportionately to the distance over which the ramp operates.

If the initial temperature profile ramps over a (dimensional) distance that is fixed at a representative pore size of 10 µm, simulated maximum pressures converge rapidly as the smallest mesh reduces through 1 µm, so that the pressures we present in figure 4 are accurate to within 0.1 bar. We note that the size of one pore is less than the size of a representative elementary volume, so that we are technically exploring beyond the limits of the continuum approach used here.

The dimensionless time range within which maximum pressure is realized is $10^{-8}$ to $10^{-6}$, corresponding to a dimensional range of about $10^{-7}$ to $10^{-5}$ s, indicating consistency with the assumption that viscous flow effects can be neglected, since viscous flow effects take more than a second to be significant.

These results raise mathematical questions, irrespective of any geophysical (textural) indications of a practical smallest mesh size or representative elementary volume, about what determines the time taken to reach the maximum pressure at the flashing front, and whether the maximum pressure value seen in figure 5 is approaching some limit as the ramp approaches a step (as the smallest mesh size approaches zero). We now derive an upper bound on the maximum pressure. We find that our bound diverges as the initial temperature gradient diverges, suggesting that the pressure maximum may be theoretically unbounded in this limit.

## 6. Maximum pressure

We seek an approximation to the maximum pressure, in the form of a formula that relates it to the material properties of the magma and the enclosed slurry. Such a formula was obtained previously [12] by using the steady-state pressure behaviour to obtain an upper bound, and

was possible only because the temperature field was approximated as constant. The present model makes it clear that temperature gradient changes are significant, and drive boiling, so it is important to properly account for the effect of early high-temperature gradients at the flashing front on the maximum pressure developed by heating.

## (a) Upper bound

We begin by deriving an upper bound on the maximum pressure developed at early times at the flashing front. We consider the reduced equation (4.8) for steam pressure or density, driven by the gradient of the temperature on the magma side of flash. We ignore the relatively small contribution from the temperature gradient on the slurry side, and movement of the flashing front is ignored.

Initial temperature in the magma is taken to be a similarity solution that starts at an earlier time $t = -t_e < 0$, so that equation (4.21) is modified to read

$$\frac{\partial T}{\partial r} = \frac{T_f - T_0}{\sqrt{\pi \epsilon_3 (t + t_e)}}, \quad r = \epsilon. \tag{6.1}$$

This is the flux from an initial temperature profile that is ramped, and a step function initial profile is recovered in the limit $t_e \to 0$. Noting that the error function is close to 1 in value when its argument is 2, the distance $\Delta r_e$ over which the initial temperature ramps from the flash value to the value 1 is given by $\Delta r_e \approx 4\sqrt{\delta_5 t_e}$.

Early pressure behaviour is governed by equation (4.8), which close to the flashing front can be written in the form

$$\frac{\partial}{\partial t}\left(\frac{p}{T}\right) = \epsilon_5 \frac{\partial}{\partial r}\left(\frac{p}{T}\frac{\partial p}{\partial r}\right). \tag{6.2}$$

At early times, pressure changes due to the influx of vapour at the flashing front propagate a distance $\Delta r \approx \sqrt{\epsilon_5 t}$ into the magma, which, together with the spatially constant initial pressures in the magma, suggests the approximation at the flashing front

$$\frac{\partial}{\partial r}\left(\frac{p}{T}\frac{\partial p}{\partial r}\right) \approx -\frac{p}{T\Delta r}\frac{\partial p}{\partial r}.$$

We put these together and take advantage of the relatively slow time rate of change of temperature to write equation (6.2) as

$$\frac{\partial p}{\partial t} \approx -\sqrt{\frac{\epsilon_5}{t}}p\frac{\partial p}{\partial r}. \tag{6.3}$$

At the flashing front, we can then approximate the early time pressure behaviour by using the reduced jump conditions (4.15) and ignoring heat flow into the slurry to set

$$p\frac{\partial p}{\partial r} \approx -\frac{T(1-T)}{\epsilon_4 \mathrm{St}\sqrt{\pi \delta_5 (t + t_e)}}. \tag{6.4}$$

Substituting this into equation (6.3) gives the following approximation for early pressure changes at the flash point:

$$\frac{\partial p}{\partial t} \approx \frac{B_1}{\sqrt{t^2 + t t_e}}, \tag{6.5}$$

where $B_1 = T(1 - T)/(\epsilon_4 \mathrm{St})\sqrt{\epsilon_5/(\pi \delta_5)}$ and $T \approx 0.4$. The solution is

$$p^e \approx p_0 + B_1\left[\ln\left(2\sqrt{t^2 + t_e t} + 2t + t_e\right) - \ln(t_e)\right]. \tag{6.6}$$

The timescale $t^*$ for reaching a maximum flash pressure is estimated by calculating when $p^e$ crosses the pressure null surface at flash, where $\partial p/\partial t = 0$. We estimate this by considering the

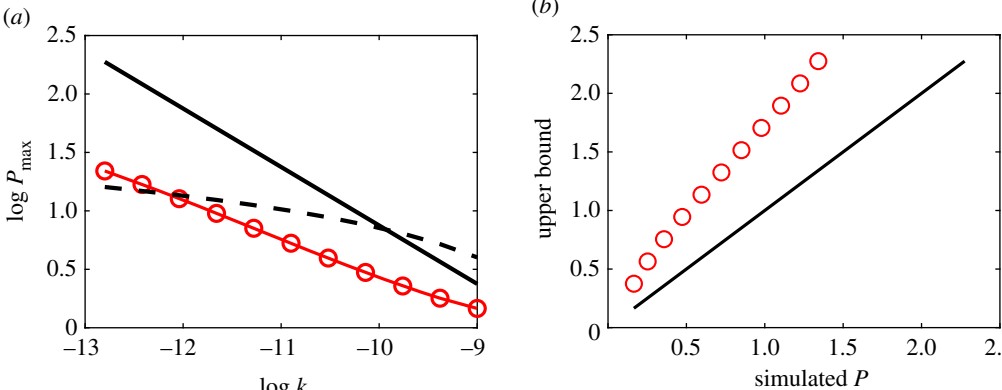

**Figure 6.** Log–log plots of simulated maximum pressures and theoretical upper bounds. Pressures are dimensionless, but are effectively in bars. Permeability values (m²) depend on porosity as in the fit (2.1) to measured values for Surtseyan bombs, with 11 porosity values chosen evenly spaced in the range [0.2, 0.8], providing the indicated values of permeability $k$. Units of permeability are m². All logarithms are to the base 10. (Online version in colour.)

density null surface, where $\partial \varrho / \partial t = 0$, since temperatures remain of order 1. Then equation (4.8) becomes the quasi-steady-state equation

$$\frac{\partial}{\partial r}\left(r^2 \frac{p}{T} \frac{\partial p}{\partial r}\right) = 0, \quad r > s(t), \tag{6.7}$$

with solution

$$p_q^2 = 2C_1 \int \frac{T}{r^2}\,\mathrm{d}r.$$

Since $T \leqslant 1$, $p_q$ has an upper bound given by $(p_q^u)^2 = 2C_1 \int r^{-2}\,\mathrm{d}r = -2C_1/r + C_2$. The boundary conditions $p(1) = 1$ and the flux condition (6.4) give $C_2 = 1 + 2C_1$, $C_1 = -\epsilon^2(1 - T)/(\epsilon_4 \mathrm{St}\sqrt{\pi}\delta_5(t + t_e))$. At $r = \epsilon$, this upper bound is closely approximated by

$$p_q^u = \frac{B_2}{(t + t_e)^{\frac{1}{4}}}, \quad B_2^2 = \frac{2\epsilon(1 - \epsilon)(1 - T)}{\epsilon_4 \mathrm{St}\sqrt{\pi}\delta_5}. \tag{6.8}$$

Dropping the term $p_0$ in equation (6.6) and equating just the leading terms in equations (6.6) and (6.8) as $t \to 0$, the time $t^*$ at which flash pressure reaches a maximum is estimated as the solution to $2B_1\sqrt{t^*/t_e} = B_2/((t_e)^{(1/4)})$, which is $t^* = B_2^2\sqrt{t_e}/(4B_1^2)$. The maximum pressure at the flashing front has an upper bound estimate of $p_{\max} = B_2 t_e^{-1/4}$, and replacing $t_e$ with its equivalent length scale $\Delta r_e$ gives $p_{\max}^2 = 8\epsilon(1 - \epsilon)(1 - T_f)/(\epsilon_4 \mathrm{St}\sqrt{\pi}\Delta r_e)$, where $T_f \approx 0.4$ is the non-dimensional flash temperature.

This can be rewritten as

$$p_{\max}^2 = \frac{8\epsilon(1 - \epsilon)(1 - T_f)T_m \mu_v \kappa_e}{\sqrt{\pi}\Delta r_e k p_a h_{\mathrm{sl}} \rho_{0s}}, \tag{6.9}$$

providing an estimate of an upper bound on the maximum pressure that develops at the flashing front, and indicating its dependence on key properties of the magma bomb.

This estimate is compared with simulated maximum pressures in figure 6, and we see that the estimated pressures are about three times the simulated values. So the estimate provides a weak upper bound for simulated maximum pressures.

## (b) Fragmentation criterion

While the upper bound illustrated in figure 6 is correct, and it is in the form of a formula, it is too weak to provide a useful fragmentation criterion for Surtseyan bombs. The log–log

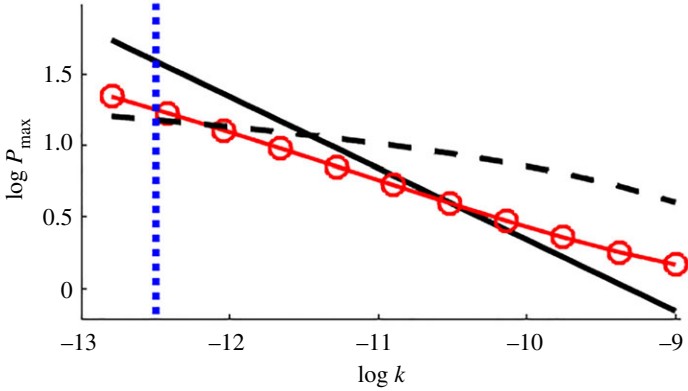

**Figure 7.** Log–log plot of simulated maximum non-dimensional pressure (line with red symbols) and the simple fitted theoretical estimate $P_{\mathrm{max}}^{\mathrm{est}}$ (equation (6.10; black line) for various permeabilities with porosities matching our data. The dashed subhorizontal black line is a representative value $20(1 - \phi)$ bars for the tensile strength of the magmatic rock when cooled. The blue vertical dashed line indicates the smallest permeability measured for intact samples of Surtseyan ejecta. Units of permeability are $m^2$. Logs are to the base 10. Other parameter values are as listed in table 1. (Online version in colour.)

graph of the upper bound in figure 6 indicates that it would be improved as an estimate of the numerical solutions by simply shifting it in log–log space. This suggests trying a simple linear approximation over the relevant range of porosities and permeabilities to relate the upper bound more closely to the more accurate numerical results. The average of numerical maximum pressure values is compared with the average of upper bound values over the range illustrated, and gives the following approximation $P_{\mathrm{max}}^{\mathrm{est}}$ to simulated pressures:

$$P_{\mathrm{max}}^{\mathrm{est}} = 0.29 p_{\mathrm{max}}, \tag{6.10}$$

where $p_{\mathrm{max}}$ is given by equation (6.9).

This approximation $P_{\mathrm{max}}^{\mathrm{est}}$ is compared with numerical values in figure 7, and with the smallest permeability $k = 0.3$ darcys observed in data from intact Surtseyan bombs. Fragmentation of a bomb corresponds to a maximum pressure that exceeds the tensile strength approximated by the dashed line in figure 7. The numerical simulations (red circles) predict fragmentation when $k < 0.6$ darcys, while the approximate formula $P_{\mathrm{max}}^{\mathrm{est}}$ (solid black line) predicts fragmentation when $k < 3$ darcys. The data in figure 3 indicate variability in measured permeability, with the smallest measured value for permeability of intact bombs in the confidence range 0.1–1.0 darcys. This contains the value $k = 0.6$ darcys, that is, the numerically predicted smallest permeability value is inside the confidence range for smallest measured permeability of intact bombs. This provides some support from field measurements for the theoretical modelling described here. There is significant uncertainty in the assumed value of tensile strength, which also affects the minimum permeability predictions for intact bombs. Note that no model fitting to data has been conducted here.

# 7. Discussion

The estimate $P_{\mathrm{max}}^{\mathrm{est}}$ is a simple formula that approximates the numerical values of maximum pressure in a Surtseyan bomb. This provides us with a fragmentation criterion that a Surtseyan bomb should fragment if the non-dimensional pressure exceeds the non-dimensional tensile strength $20(1 - \phi)$, that is, combining equations (6.10) and (6.9), if

$$P_{\mathrm{max}}^{\mathrm{est}} = \sqrt{\frac{0.7\epsilon(1 - \epsilon)(1 - T_f)T_m \mu_v \kappa_e}{\sqrt{\pi}\,\Delta r_e k\, p_a h_{\mathrm{sl}} \rho_{0s}}} > 20(1 - \phi). \tag{7.1}$$

We can compare our fragmentation criterion (7.1) with the dimensional estimate for maximum pressure $p_{\max}^{\mathrm{old}} = \sqrt{(7(1 - \epsilon)(T_m - T_0)\mu_v\kappa_e/kh_{\mathrm{sl}})(RT_m/M)}$ obtained in earlier work [12] by writing our result in equation (7.1) in terms of the dimensional value for pressure

$$p_{\max}^{\mathrm{est}} = \sqrt{\frac{0.7(1 - \epsilon)(1 - T_f)T_m\mu_v\kappa_e}{\sqrt{\pi}kh_{\mathrm{sl}}} \left( \frac{RT_m}{M} \right) \left( \frac{\epsilon}{\Delta r} \right)}\mathrm{Pa} \approx \sqrt{\frac{0.1\epsilon}{\Delta r}}p_{\max}^{\mathrm{old}}. \qquad (7.2)$$

The form of $p_{\max}^{\mathrm{old}}$ differs from $p_{\max}^{\mathrm{est}}$ most significantly in that our new result has an extra factor $\approx \sqrt{0.1\epsilon/\Delta r}$, involving the ratio of the inclusion size to the ramping distance for the assumed initial temperature profile. This extra factor means that our present estimate of minimum permeability of 0.6 darcys for a bomb to survive in flight is significantly higher than the estimate of 0.02 darcys in [12]. This previous estimate is well below the smallest permeability (0.1–1.0 darcys) measured in intact bombs, while our new result lies inside this range. In this sense, our present model is more consistent with the data from intact bombs, since the previous result says that some intact bombs would be expected to have permeabilities below the measured smallest values 0.1–1.0 darcys.

The term $\Delta r$ is a measure of the size of the region over which initial temperature ramps from boiling values at the flashing front up to magma values. In the limit as it goes to zero, and the initial temperature profile approaches a step function, our upper bound tends to infinity. Our approximate early time solution (6.6) also grows without bound in this limit, suggesting our model may be mathematically ill-posed if it is assumed that the initial temperature profile is a step function. Such an assumption is common for the initial temperature profiles in heat conduction problems, where, owing to the stable behaviour of solutions to diffusion equations, it does not cause any problems. The ill-posedness might be regularized by re-scaling pressure to include previously neglected small parameters such as the pressure–work term in the $\dot{s}$ equation just before equation (4.15), or by recognizing the small distance over which initial flash pressures change and hence including viscous dissipation terms.

However, a step function initial temperature profile is not physically realistic, and a ramped profile already provides us with a mathematically better behaved model. In a volcanological context, the length scale $\Delta r$ might be considered to be the size of a representative elementary volume, which would be several times the mean pore size.

The measurements made on intact bombs shown in figure 3 suggest minimum permeabilities in the range 0.1–1.0 darcys. This is broadly consistent with our model results. We note that this is a one-sided view of all ejected bombs, applying only to intact bombs. While it is consistent with the hypothesis that bombs with smaller permeability will fragment, there are no measurements available for fragmented bombs to confirm this. Even if there were such measurements, we would have no assurance of the cause of fragmentation. Fragmentation due to impact with the ground is not modelled here—we consider only the mechanism of steam generation and the associated steam pressures. Given the highly vesicular nature of the ejected magma, it is possible that all ejected bombs have higher permeabilities than the critical value for fragmentation. If so, our model then provides a physical explanation for why typical Surtseyan ejecta are not expected to fragment as a result of steam pressure build-up from flashing of slurry inclusions after ejection, in that the vapour typically escapes before pressures reach fragmentation values.

## 8. Conclusion

We have developed and simplified a fully transient sphere-within-a-sphere model for the pressure increase expected to occur inside Surtseyan bombs after ejection, owing to flashing to steam of liquid in slurry inclusions. We reduced our results to a single formula (7.2) for the maximum pressure developed at the flashing front, revealing how that pressure depends on permeability and relative size of the inclusion. This formula, when compared with the tensile strength of the bomb, provides a new fragmentation criterion for Surstseyan bombs.

Our model neglects the effect of rotation, which we expect to assist steam transport and reduce the maximum pressure if included in the model. The thermal capacity of magma has been approximated as a constant value, ignoring its variation of $\pm 30\%$ over the temperature range modelled. The geometry of bombs is not spherical, and multiple inclusions are found inside each one at varying distances from the surface of the bomb. The deformation of the porous matrix near slurry inclusions that is observed in samples suggests that a more sophisticated approach like that in [13] might be useful, explicitly considering stress and strain in the rock matrix. Our work already shows that pressure changes due to flashing will propagate more rapidly than temperature changes, relevant to questions about compression effects near inclusions. A region of reduced permeability in the matrix next to an inclusion may also be a useful future modification of our model.

The mechanism for insertion of slurry into the hot magma immediately prior to ejection is important for initial temperature profiles, and is a more complicated flow and heat transport problem that may lead to estimates of $\Delta r$ that better reflect the volcanology of Surtseyan eruptions.

Our results highlight the importance of the initial temperature gradient in the hot magma adjacent to the flashing front, driving boiling and hence pressure increases there. Numerical simulations illustrated in figure 4 using a ramping distance for the initial temperature that is given by a typical pore size indicate that fragmentation is expected to occur for any bombs with permeabilities less that about 1 darcy. For these relatively low-permeability bombs, steam pressure build-up due to heating by surrounding hot magma is not adequately relieved by steam escape through the porous magma. This, together with the observation that all measured permeabilities of intact ejecta are greater than 1 darcy, provides an explanation for why most Surtseyan bombs survive steam pressures developed inside as a result of flashing of slurry inclusions.

Data accessibility. Data are provided as electronic supplementary material. These are original data provided by C.I.S.

Authors' contributions. E.G. and M.J.M. contributed equally to the mathematical modelling, asymptotics and numerical simulations; C.I.S. provided all data, the field work and data sections and most of the volcanological expertise, together with the fundamental fragmentation question that motivated this work. All authors wrote/revised the paper, approved the final version and agreed to be accountable for all aspects of the work.

Competing interests. We declare we have no competing interests.

Funding. No funding has been received for this article.

Acknowledgements. We are grateful to Andrew Fowler (University of Limerick and University of Oxford) for fruitful discussions during the early development of our model.

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
