## [Peer Review File · Proceedings. Mathematical, Physical, and Engineering Sciences]

Review History

RSPA-2021-0166.R0 (Original submission)

Review form: Referee 1

Is the manuscript an original and important contribution to its field?

Acceptable

Is the paper of sufficient general interest?

Acceptable

Is the overall quality of the paper suitable?

Good

Can the paper be shortened without overall detriment to the main message?

Yes

Do you think some of the material would be more appropriate as an electronic appendix?

Yes

Do you have any ethical concerns with this paper?

No

Recommendation?

Major revision is needed (please make suggestions in comments)

Comments to the Author(s)

Review of RSPA-2021-0166 on 6 April 2021

The manuscript under review elaborates a mathematical model for Surtseyan bombs, which are a particular type of volcanic ejecta where hot vesicular lava encloses a cool, water saturated chunk of tephra. The manuscript reports microstructure measurements that are used to estimate the permeability, the development of a mathematical model based on physical arguments, the analysis of that model, and results of numerical simulations and analytical approximations. A previous paper by the same authors developed the original model, which made the simplifying approximation of a frozen temperature field. In the present work, the authors couple the temperature and pressure fields. They obtain a result that differs by a ratio of length scales associated with the initial temperature condition, and is potentially large. They conclude by noticing that their model predicts a minimum permeability to avoid fracture, and that there are no observations of bombs with permeability below this minimum.

The manuscript has a lot of nice features, including novel permeability measurements, an appealing physical model that makes reasonable idealisations, and clever analysis in estimating the relevant physical scales and making simplifications. However it also has some significant problems that should be addressed before I can recommend publication.

The first of these is the cursory introduction that doesn't provide a clear motivation for the paper. It gives a brief overview of the physical picture of Surtseyan bombs, but doesn't discuss a motivation for physical/mathematical modelling of these features. This might have taken the form of questions to be addressed and the hypothesis that is inherent in the models (the pressure of flashing is relieved by porous flow of steam when the permeability is large enough). Moreover, the introduction doesn't discuss why previous theoretical work is inadequate in addressing the questions. Finally, the introduction should provide an overview of the manuscript, explaining its organisation and, I suggest, also its key results. Instead the introduction breaks abruptly at a subsection called "Methods" which is really just about the methods of the permeability measurements (and the results) and then comes back to provide some description of the model. The text and figure about permeability measurements should be moved out of the introduction and into their own section, afterwards.

The manuscript also lacks a more detailed discussion of the model and its limitations, the results and their implications. What discussion there is provided in a very brief conclusion section. The last two sentences seem to be the only reference back to the motivating questions (presumably) that connect the mathematics to the physical volcanology. I discuss my sense of the model limitations below. But there is a lack of connection to the physical volcanology here. What can equation (6.1) and Figure 5 tell us about the eruption process and materials? If nothing, why?

The model development in section 2 seems a reasonable balance between explaining the physics (though see detailed points below) and not getting too bogged down in the details that will later be neglected. The model reduction section, however, seems to move hastily through a series of approximations based on the size of dimensionless parameters, dropping terms without discussion of the physics that they represent. This seems problematic in that the initial conditions then create transients in which some variables (and their derivatives) are much larger than unity. Because of the lack of physical discussion, it is difficult to see which of the neglected terms might become important, and what its qualitative effect would be. While it is reasonable to make simplifications, the authors provide no discussion a posteriori of the consequences of these simplifications, and hence of the limitations of the results.

Two simplifications that concern me are as follows. In eqn (3.8), the term in δ_3 is neglected. δ_3 is indeed small, but near the flashing front (where all the interesting action occurs), $\partial p / \partial t$ can be very large. In equation (3.13), the term from the previous equation in $\lambda_2 \epsilon_4$ has been neglected relative to the jump in heat flux. There is only a factor

of 10 in the coefficients, but p can be very large near the flashing front. So in both cases (and perhaps others) it seems somewhat unclear whether neglecting the terms can be justified on physical grounds. At the very least, some discussion of this a posteriori should be included.

A lot of attention is then given to the nifty matched asymptotics for the analytical approximation, which in Fig. 3 seems to do a reasonable job of matching some aspects of the solution. But it seems to fail badly in matching the maximum pressure, which is the key physical output of interest. So after reaching figure 5, it is unclear why the asymptotic model is included in so much detail (or at all). What do we learn from it of value? In figure 5a, the mismatch between numerics and the approximation is as big as the difference between the fracture criterion and the numerics. It feels like much of the manuscript is driven by an interest in mathematical analysis rather than an interest in the physical predictions of the model. I find this problematic, given the motivation of the work.

I am rather dubious of the implication that the model is correct because of the lack of observations of permeabilities less than one Darcy. One Darcy is very small for a magma with porosity of 0.35--0.8, even if the pores are bubbles. The measurements were all made on bombs containing inclusions, I think, which is a sampling bias. What if fragments of fractured bombs had permeabilities in the same range?

Detailed comments by page:line(s). (Page numbers are as in the header, not the black box)

3:13. How are "bombs" related to "lump of lava" and "tail of tephra"? Are they the lump?

3:56. What is "Palabos"?

4:12. Can this be described using a more typical form such as Kozeny-Carman? Is this form usual or just convenient?

4:52. So the model must be accurate over times of less than a second. This suggests that the choice of initial condition is paramount. Obviously getting the IC right isn't easy...

6:29. It is a bit hard to see how the averaging was conducted. Is the microscopic pressure equal to the continuum pressure? What about the stress τ ? The usual upscaling would give a dissipation rate of something like $K_e u_l^2$ for laminar flow (but you haven't yet assumed laminar flow). I think it would be better to jump right to the continuum equations.

(2.5). A better choice of notation to disambiguate thermal conductivity and permeability would be nice, while avoiding K_{el} , which looks more like a bulk modulus.

(2.8). Is the β in this equation the same as the β in line 57 of the same page? I think they're different...

6:57. "this equation" -- indefinite antecedent.

(2.9). Again the dissipation term is problematic.

9:29. "over-estimate" by how much? What effect does this have on other approximations made to simplify the equations?

11:52 (and elsewhere). The Ideal Gas law, not the perfect gas equation?

13:11. Mention use of subscript notation for partial derivatives

Fig 3. Needs panel labels and a much clearer caption that refers to the panels. Maybe break into two figures? Needs much more explanation in the text.

15:10. If the mesh size that is needed on the basis of characteristics of the solution is similar to the pore size, then there's a potential problem with the continuum assumptions. This needs to be highlighted.

15:58. typo

17:47. "they" -- indefinite antecedent. Confusing sentence.

17:58. It is concerning that the solution for maximum pressure depends on a ratio that can be arbitrarily large. First of all, it is unrealistic to assume that the enclosure is captured in such a way that there is a step-function in temperature. And secondly, if the mathematics doesn't regularise this then I think the model is ill-posed (possibly due to neglecting a "small" term that isn't small near the flashing front at small time).

Review form: Referee 2

Is the manuscript an original and important contribution to its field?

Good

Is the paper of sufficient general interest?

Marginal

Is the overall quality of the paper suitable?

Acceptable

Can the paper be shortened without overall detriment to the main message?

No

Do you think some of the material would be more appropriate as an electronic appendix?

Yes

Do you have any ethical concerns with this paper?

No

Recommendation?

Major revision is needed (please make suggestions in comments)

Comments to the Author(s)

Thank you for an interesting paper to review and apologies for my lateness.

You have presented some very detailed models of magma-slurry interaction in Surtseyan eruptions, and consider a range of factors that include the permeable escape of vapor and the potential failure of bombs due to pressure accumulation. The questions that you are asking **are** important, and the overall objectives of the paper could be more clearly communicated to better reflect this.

As you will see from the attached annotated .pdf of the paper, I have added some comments and questions to the text. These largely reflect my encouragement for you to build a much stronger bridge with the existing literature, (re)consider some aspects of the modelling, and add a discussion section in which you can emphasize the importance of your model results to our understanding of Surtseyan eruptions.

My major comments/criticisms are that:

1. Stronger bridges are needed with existing literature on magma-water interactions. Yes, it is an interesting thought experiment to explore whether a Surtseyan bomb will explode due to the high pressure of vaporized entrained steam, but could you please consider and emphasize the importance of this process? How might the physics of magma-water interaction be affected? And the eruption mechanisms?
2. Please give more information about the geological setting of the studied bombs, including the deposits, bomb sizes, and inferred emplacement mechanism. It is stated that the studied bombs were smaller than typical bombs at the study site, but more information is needed.
3. A cartoon illustrating processes in Surtseyan eruptions will greatly improve the paper. This could show the water body, vent, crater, slurry, etc, and also zoom in to show small-scale magma-slurry interactions being considered in the models.
4. Please discuss whether the vesicularity of bombs is entirely pre-fragmentation, or whether some post-fragmentation vesiculation also occurs. Do the current bomb textures

faithfully record the moment of slurry ingress? If not, then should the final permeability values be treated with some caution?

5. Please consider the thermal and physical state of the “host” bomb close to the slurry. Was it above or below the glass transition temperature when the vapor expansion occurred? It was stated that the near-slurry magma looked compressed, and then stated that compression of magma was not considered. I think it is important to consider and discuss whether the slurry vapor expanded when the bomb was still hot enough to viscously deform. If that had happened, then the vapor expansion and pressurisation could hypothetically drive compaction and collapse of the vesicles.
6. It would be useful to explicitly state whether you consider the pressure dependence of the vaporization temperature of water.
7. The discussion of mesh size vs model results is somewhat concerning as it shows to what an extent the model set-up affects the outputs. I'd like to see a deeper discussion of this, and the relationship with the vesicle size, which is touched upon. Some of this writing on model set-up could fit best in an appendix.
8. Have you considered what particle size is present in the slurry? The Fig. 1 image suggests only large clasts, but we know that there can be motion of very small fragments of basaltic magma through fractures and pores in basalt (Owen et al 2019 JVGR, Taddeucci et al 2021 Nat Geoscience). Do you see any sign of small particles having moved through the connected bubbles (and perhaps welded)?
9. Please add a discussion section, in which you discuss the relevance of the model results, their implications for physical processes occurring in Surtseyan eruptions, the shortcomings of the modelling approach, prospect for future work, and more. Stronger connection with existing literature would be useful here.
10. The conclusion is a real anti-climax, suggesting that there are only modest advances made on a simpler version of the model published elsewhere. Please reconsider your tone here!
11. Figures need attention – please add (a), (b) (c) etc., and see specific comments written directly on the manuscript. The schematic cartoons are a must.

Please see the attached manuscript for many more comments.

Best wishes,
Hugh Tuffen

Decision letter (RSPA-2021-0166.R0)

04-May-2021

Dear Dr Greenbank

The Editor of Proceedings A has now received comments from referees on the above paper and would like you to revise it in accordance with their suggestions which can be found below (not including confidential reports to the Editor).

Please submit a copy of your revised paper within four weeks - if we do not hear from you within this time then it will be assumed that the paper has been withdrawn. In exceptional circumstances, extensions may be possible if agreed with the Editorial Office in advance.

Please note that it is the editorial policy of Proceedings A to offer authors one round of revision in which to address changes requested by referees. If the revisions are not considered satisfactory by the Editor, then the paper will be rejected, and not considered further for publication by the journal. In the event that the author chooses not to address a referee's comments, and no scientific justification is included in their cover letter for this omission, it is at the discretion of the Editor whether to continue considering the manuscript.

To revise your manuscript, log into <http://mc.manuscriptcentral.com/prsa> and enter your Author Centre, where you will find your manuscript title listed under "Manuscripts with Decisions." Under "Actions," click on "Create a Revision." Your manuscript number has been appended to denote a revision.

You will be unable to make your revisions on the originally submitted version of the manuscript. Instead, revise your manuscript and upload a new version through your Author Centre.

When submitting your revised manuscript, you will be able to respond to the comments made by the referee(s) and upload a file "Response to Referees" in Step 1: "View and Respond to Decision Letter". Please use this to document how you have responded to the comments, and the adjustments you have made. In order to expedite the processing of the revised manuscript, please be as specific as possible in your response to the referee(s).

IMPORTANT: Your original files are available to you when you upload your revised manuscript. Please delete any unnecessary previous files before uploading your revised version.

When revising your paper please ensure that it remains under 28 pages long. In addition, any pages over 20 will be subject to a charge (£150 + VAT (where applicable) per page). Your paper has been ESTIMATED to be 18 pages pages.

Open Access

You are invited to opt for open access, our author pays publishing model. Payment of open access fees will enable your article to be made freely available via the Royal Society website as soon as it is ready for publication. For more information about open access please visit <https://royalsociety.org/journals/authors/open-access/>. The open access fee for this journal is £1700/\$2380/€2040 per article. VAT will be charged where applicable. Please note that if the corresponding author is at an institution that is part of a Read and Publishing deal you are required to select this option. See <https://royalsociety.org/journals/librarians/purchasing/read-and-publish/read-publish-agreements/> for further details.

Once again, thank you for submitting your manuscript to Proc. R. Soc. A and I look forward to receiving your revision. If you have any questions at all, please do not hesitate to get in touch.

Yours sincerely
Raminder Shergill
proceedingsa@royalsociety.org

on behalf of
Professor Colin Meyer
Board Member
Proceedings A

Reviewer(s)' Comments to Author:
Referee: 1
Comments to the Author(s)
Review of RSPA-2021-0166 on 6 April 2021

The manuscript under review elaborates a mathematical model for Surtseyan bombs, which are a particular type of volcanic ejecta where hot vesicular lava encloses a cool, water saturated chunk of tephra. The manuscript reports microstructure measurements that are used to estimate the permeability, the development of a mathematical model based on physical arguments, the analysis of that model, and results of numerical simulations and analytical approximations. A previous paper by the same authors developed the original model, which made the simplifying approximation of a frozen temperature field. In the present work, the authors couple the temperature and pressure fields. They obtain a result that differs by a ratio of length scales associated with the initial temperature condition, and is potentially large. They conclude by noticing that their model predicts a minimum permeability to avoid fracture, and that there are no observations of bombs with permeability below this minimum.

The manuscript has a lot of nice features, including novel permeability measurements, an appealing physical model that makes reasonable idealisations, and clever analysis in estimating the relevant physical scales and making simplifications. However it also has some significant problems that should be addressed before I can recommend publication.

The first of these is the cursory introduction that doesn't provide a clear motivation for the paper. It gives a brief overview of the physical picture of Surtseyan bombs, but doesn't discuss a motivation for physical/mathematical modelling of these features. This might have taken the form of questions to be addressed and the hypothesis that is inherent in the models (the pressure of flashing is relieved by porous flow of steam when the permeability is large enough). Moreover, the introduction doesn't discuss why previous theoretical work is inadequate in addressing the questions. Finally, the introduction should provide an overview of the manuscript, explaining its organisation and, I suggest, also its key results. Instead the introduction breaks abruptly at a subsection called "Methods" which is really just about the methods of the permeability measurements (and the results) and then comes back to provide some description of the model. The text and figure about permeability measurements should be moved out of the introduction and into their own section, afterwards.

The manuscript also lacks a more detailed discussion of the model and its limitations, the results and their implications. What discussion there is provided in a very brief conclusion section.

The last two sentences seem to be the only reference back to the motivating questions (presumably) that connect the mathematics to the physical volcanology. I discuss my sense of the model limitations below. But there is a lack of connection to the physical volcanology here. What can equation (6.1) and Figure 5 tell us about the eruption process and materials? If nothing, why?

The model development in section 2 seems a reasonable balance between explaining the physics (though see detailed points below) and not getting too bogged down in the details that will later be neglected. The model reduction section, however, seems to move hastily through a series of approximations based on the size of dimensionless parameters, dropping terms without discussion of the physics that they represent. This seems problematic in that the initial conditions then create transients in which some variables (and their derivatives) are much larger than unity. Because of the lack of physical discussion, it is difficult to see which of the neglected terms might become important, and what its qualitative effect would be. While it is reasonable to make simplifications, the authors provide no discussion a posteriori of the consequences of these simplifications, and hence of the limitations of the results.

Two simplifications that concern me are as follows. In eqn (3.8), the term in δ_3 is neglected. δ_3 is indeed small, but near the flashing front (where all the interesting action occurs), $\partial p / \partial t$ can be very large. In equation (3.13), the term from the previous equation in $\lambda_2 \epsilon_4$ has been neglected relative to the jump in heat flux. There is only a factor of 10 in the coefficients, but p can be very large near the flashing front. So in both cases (and perhaps others) it seems somewhat unclear whether neglecting the terms can be justified on physical grounds. At the very least, some discussion of this a posteriori should be included.

A lot of attention is then given to the nifty matched asymptotics for the analytical approximation, which in Fig. 3 seems to do a reasonable job of matching some aspects of the solution. But it seems to fail badly in matching the maximum pressure, which is the key physical output of interest. So after reaching figure 5, it is unclear why the asymptotic model is included in so much detail (or at all). What do we learn from it of value? In figure 5a, the mismatch between numerics and the approximation is as big as the difference between the fracture criterion and the numerics. It feels like much of the manuscript is driven by an interest in mathematical analysis rather than an interest in the physical predictions of the model. I find this problematic, given the motivation of the work.

I am rather dubious of the implication that the model is correct because of the lack of observations of permeabilities less than one Darcy. One Darcy is very small for a magma with porosity of 0.35--0.8, even if the pores are bubbles. The measurements were all made on bombs containing inclusions, I think, which is a sampling bias. What if fragments of fractured bombs had permeabilities in the same range?

Detailed comments by page:line(s). (Page numbers are as in the header, not the black box)

3:13. How are "bombs" related to "lump of lava" and "tail of tephra"? Are they the lump?

3:56. What is "Palabos"?

4:12. Can this be described using a more typical form such as Kozeny-Carman? Is this form usual or just convenient?

4:52. So the model must be accurate over times of less than a second. This suggests that the choice of initial condition is paramount. Obviously getting the IC right isn't easy...

6:29. It is a bit hard to see how the averaging was conducted. Is the microscopic pressure equal to the continuum pressure? What about the stress τ ? The usual upscaling would give a dissipation rate of something like $K_e u_l^2$ for laminar flow (but you haven't yet assumed laminar flow). I think it would be better to jump right to the continuum equations.

(2.5). A better choice of notation to disambiguate thermal conductivity and permeability would be nice, while avoiding K_{el} , which looks more like a bulk modulus.

(2.8). Is the β in this equation the same as the β in line 57 of the same page? I think they're different...

6:57. "this equation" -- indefinite antecedent.

(2.9). Again the dissipation term is problematic.

9:29. "over-estimate" by how much? What effect does this have on other approximations made to simplify the equations?

11:52 (and elsewhere). The Ideal Gas law, not the perfect gas equation?

13:11 Mention use of subscript notation for partial derivatives

Fig 3. Needs panel labels and a much clearer caption that refers to the panels. Maybe break into two figures? Needs much more explanation in the text.

15:10. If the mesh size that is needed on the basis of characteristics of the solution is similar to the pore size, then there's a potential problem with the continuum assumptions. This needs to be highlighted.

15:58. typo

17:47. "they" -- indefinite antecedent. Confusing sentence.

17:58. It is concerning that the solution for maximum pressure depends on a ratio that can be arbitrarily large. First of all, it is unrealistic to assume that the enclosure is captured in such a way that there is a step-function in temperature. And secondly, if the mathematics doesn't regularise this then I think the model is ill-posed (possibly due to neglecting a "small" term that isn't small near the flashing front at small time).

Referee: 2

Comments to the Author(s)

Thank you for an interesting paper to review and apologies for my lateness.

You have presented some very detailed models of magma-slurry interaction in Surtseyan eruptions, and consider a range of factors that include the permeable escape of vapor and the potential failure of bombs due to pressure accumulation. The questions that you are asking *are* important, and the overall objectives of the paper could be more clearly communicated to better reflect this.

As you will see from the attached annotated .pdf of the paper, I have added some comments and questions to the text. These largely reflect my encouragement for you to build a much stronger bridge with the existing literature, (re)consider some aspects of the modelling, and add a discussion section in which you can emphasize the importance of your model results to our understanding of Surtseyan eruptions.

My major comments/criticisms are that:

1. Stronger bridges are needed with existing literature on magma-water interactions. Yes, it is an interesting thought experiment to explore whether a Surtseyan bomb will explode due to the high pressure of vaporized entrained steam, but could you please consider and emphasize the importance of this process? How might the physics of magma-water interaction be affected? And the eruption mechanisms?
2. Please give more information about the geological setting of the studied bombs, including the deposits, bomb sizes, and inferred emplacement mechanism. It is stated that the studied bombs were smaller than typical bombs at the study site, but more information is needed.
3. A cartoon illustrating processes in Surtseyan eruptions will greatly improve the paper. This could show the water body, vent, crater, slurry, etc, and also zoom in to show small-scale magma-slurry interactions being considered in the models.
4. Please discuss whether the vesicularity of bombs is entirely pre-fragmentation, or whether some post-fragmentation vesiculation also occurs. Do the current bomb textures faithfully record the moment of slurry ingress? If not, then should the final permeability values be treated with some caution?
5. Please consider the thermal and physical state of the "host" bomb close to the slurry. Was it above or below the glass transition temperature when the vapor expansion occurred? It was stated that the near-slurry magma looked compressed, and then stated that compression of magma was not considered. I think it is important to consider and discuss whether the slurry vapor expanded when the bomb was still hot enough to viscously deform. If that had happened, then the vapor expansion and pressurisation could hypothetically drive compaction and collapse of the vesicles.
6. It would be useful to explicitly state whether you consider the pressure dependence of the vaporization temperature of water.
7. The discussion of mesh size vs model results is somewhat concerning as it shows to what an extent the model set-up affects the outputs. I'd like to see a deeper discussion of this, and the relationship with the vesicle size, which is touched upon. Some of this writing on model set-up could fit best in an appendix.
8. Have you considered what particle size is present in the slurry? The Fig. 1 image suggests only large clasts, but we know that there can be motion of very small fragments of basaltic magma through fractures and pores in basalt (Owen et al 2019 JVGR, Taddeucci et al 2021 Nat Geoscience). Do you see any sign of small particles having moved through the connected bubbles (and perhaps welded)?
9. Please add a discussion section, in which you discuss the relevance of the model results, their implications for physical processes occurring in Surtseyan eruptions, the shortcomings of the

modelling approach, prospect for future work, and more. Stronger connection with existing literature would be useful here.

10. The conclusion is a real anti-climax, suggesting that there are only modest advances made on a simpler version of the model published elsewhere. Please reconsider your tone here!

11. Figures need attention – please add (a), (b) (c) etc., and see specific comments written directly on the manuscript. The schematic cartoons are a must.

Please see the attached manuscript for many more comments.

Best wishes,
Hugh Tuffen

Board Member:

Comments to Author(s):

The two detailed review reports suggest major revisions and both recommend a deeper connection with the literature.

Board member pre-assessment comments (if available):

This looks like an interesting and potentially important paper on Surtseyan eruptions, a type of basaltic eruptions. I am excited to hear what the reviewers think.

RSPA-2021-0166.R1 (Revision)

Review form: Referee 1

Is the manuscript an original and important contribution to its field?

Good

Is the paper of sufficient general interest?

Acceptable

Is the overall quality of the paper suitable?

Good

Can the paper be shortened without overall detriment to the main message?

Yes

Do you have any ethical concerns with this paper?

No

Recommendation?

Accept with minor revision (please list in comments)

Comments to the Author(s)

The authors have done a nice job of revising the manuscript and responding to reviewer comments. I think their a posteriori checks on the insignificance of terms is an important confirmation of the approximations.

The writing in the manuscript could be better; there are various places where sentences are convoluted and/or too long, or the wording is awkward or repetitive.

Although it is probably inconsequential, their representation of dissipation by Darcy flow is wrong, as I think I mentioned in my previous review. I refer to the unnumbered equation after (3.11), which uses the viscosity times the square of what the authors seem to consider a strain rate. The actual dissipation in porous flow occurs in the Poiseuille-type flow in the pore-throats, at a strain-rate that is something like the microscopic flow speed divided by the pore-throat radius. At the continuum scale, this is given by $\mu/k * u.u$, where μ is viscosity, k is permeability, and $u.u$ is the square of the magnitude of the Darcy flux vector. I'm sure this is in a standard text such as Bear's *Dynamics of Fluids in Porous Media*.

comments by page-line

4-18: Description of the Deborah number unclear. Deform what? viscous relaxation of what?

4-35-40: sentence is way too long and complex

4-43: Methods seems like the wrong section heading name.

4-50: "universally composites" -- awkward phrasing

6-20: "acknowledges a posteriori" -- awkward phrasing. Do you mean a priori?

7-17: "that inclusions have similar properties to surrounding rock" -- this needs a citation and a clearer explanation of what "similar" means.

7-47: See note above; I think your formulation of Darcy dissipation is incorrect.

9-45: Units on the dimensional temperature of 300?

10-7: "expected to be weaker" -- actually I would say that the $(1-\phi)$ factor accounts for the fractional area that is load-bearing.

10-13: "restrict to spherical geometry" -- wasn't this already done above?

16-56: "We will derive an upper..." -- this sentence seems to make no sense (though I think I understand that you are trying to say that the upper bound diverges as the gradient diverges)

18-57: "the smallest data value for permeability of intact bombs..." -- this is an important sentence but the logic is not sufficiently clear. Be explicit about the nature of the "consistency".

(7.2): write this as a multiple of p_{max}^{old} ?

20-38: only a mathematician would use the "representative elementary volume" as a lengthscale "in a volcanological context"! It is a convenient mathematical construct!

Decision letter (RSPA-2021-0166.R1)

27-Jul-2021

Dear Dr Greenbank,

On behalf of the Editor, I am pleased to inform you that your Manuscript RSPA-2021-0166.R1 entitled "A theoretical model of Surtseyan bomb fragmentation" has been accepted for publication subject to minor revisions in Proceedings A. Please find the referees' comments below.

The reviewer(s) have recommended publication, but also suggest some minor revisions to your manuscript. Therefore, I invite you to respond to the reviewer(s)' comments and revise your manuscript. Please note that we have a strict upper limit of 28 pages for each paper. Please endeavour to incorporate any revisions while keeping the paper within journal limits. Please note that page charges are made on all papers longer than 20 pages. If you cannot pay these charges you must reduce your paper to 20 pages before submitting your revision. Your paper has been ESTIMATED to be 22 pages. We cannot proceed with typesetting your paper without your agreement to meet page charges in full should the paper exceed 20 pages when typeset. If you have any questions, please do get in touch.

It is a condition of publication that you submit the revised version of your manuscript within 7 days. If you do not think you will be able to meet this date please let me know in advance of the due date.

To revise your manuscript, log into <https://mc.manuscriptcentral.com/prsa> and enter your Author Centre, where you will find your manuscript title listed under "Manuscripts with Decisions." Under "Actions," click on "Create a Revision." Your manuscript number has been appended to denote a revision.

You will be unable to make your revisions on the originally submitted version of the manuscript. Instead, revise your manuscript and upload a new version through your Author Centre.

When submitting your revised manuscript, you will be able to respond to the comments made by the referee(s) and upload a file "Response to Referees" in Step 1: "View and Respond to Decision Letter". Please provide a point-by-point response to the comments raised by the reviewers and the editor(s). A thorough response to these points will help us to assess your revision quickly. You can also upload a 'tracked changes' version either as part of the 'Response to reviews' or as a 'Main document'.

IMPORTANT: Your original files are available to you when you upload your revised manuscript. Please delete any redundant files before completing the submission process.

When uploading your revised files, please make sure that you include the following as we cannot proceed without these:

- 1) A text file of the manuscript (doc, txt, rtf or tex), including the references, tables (including captions) and figure captions. Please remove any tracked changes from the text before submission. PDF files are not an accepted format for the "Main Document".
- 2) A separate electronic file of each figure (tif, eps or print-quality pdf preferred). The format should be produced directly from original creation package, or original software format.
- 3) Electronic Supplementary Material (ESM): all supplementary materials accompanying an accepted article will be treated as in their final form. Note that the Royal Society will not edit or typeset supplementary material and it will be hosted as provided. Please ensure that the supplementary material includes the paper details where possible (authors, article title, journal name). Supplementary files will be published alongside the paper on the journal website and posted on the online figshare repository (<https://figshare.com>). The heading and legend provided for each supplementary file during the submission process will be used to create the figshare page, so please ensure these are accurate and informative so that your files can be found in searches. Files on figshare will be made available approximately one week before the accompanying article so that the supplementary material can be attributed a unique DOI.

Alternatively you may upload a zip folder containing all source files for your manuscript as described above with a PDF as your "Main Document". This should be the full paper as it appears when compiled from the individual files supplied in the zip folder.

Article Funder

Please ensure you fill in the Article Funder question on page 2 to ensure the correct data is collected for FundRef (<http://www.crossref.org/fundref/>).

Media summary

Please ensure you include a short non-technical summary (up to 100 words) of the key findings/importance of your paper. This will be used for to promote your work and marketing purposes (e.g. press releases). The summary should be prepared using the following guidelines:

*Write simple English: this is intended for the general public. Please explain any essential technical terms in a short and simple manner.

*Describe (a) the study (b) its key findings and (c) its implications.

*State why this work is newsworthy, be concise and do not overstate (true 'breakthroughs' are a rarity).

*Ensure that you include valid contact details for the lead author (institutional address, email address, telephone number).

Cover images

We welcome submissions of images for possible use on the cover of Proceedings A. Images should be square in dimension and please ensure that you obtain all relevant copyright permissions before submitting the image to us. If you would like to submit an image for consideration please send your image to proceedingsa@royalsociety.org

Open Access

You are invited to opt for open access, our author pays publishing model. Payment of open access fees will enable your article to be made freely available via the Royal Society website as soon as it is ready for publication. For more information about open access please visit <https://royalsociety.org/journals/authors/open-access/>. The open access fee for this journal is £1700/\$2380/€2040 per article. VAT will be charged where applicable. Please note that if the corresponding author is at an institution that is part of a Read and Publishing deal you are required to select this option. See <https://royalsociety.org/journals/librarians/purchasing/read-and-publish/read-publish-agreements/> for further details.

Once again, thank you for submitting your manuscript to Proceedings A and I look forward to receiving your revision. If you have any questions at all, please do not hesitate to get in touch.

Best wishes

Raminder Shergill

proceedingsa@royalsociety.org

Proceedings A

on behalf of

Professor Colin Meyer

Board Member

Proceedings A

Reviewer(s)' Comments to Author:

Referee: 1

Comments to the Author(s)

The authors have done a nice job of revising the manuscript and responding to reviewer comments. I think their a posteriori checks on the insignificance of terms is an important confirmation of the approximations.

The writing in the manuscript could be better; there are various places where sentences are convoluted and/or too long, or the wording is awkward or repetitive.

Although it is probably inconsequential, their representation of dissipation by Darcy flow is wrong, as I think I mentioned in my previous review. I refer to the unnumbered equation after (3.11), which uses the viscosity times the square of what the authors seem to consider a strain rate. The actual dissipation in porous flow occurs in the Poiseuille-type flow in the pore-throats, at a strain-rate that is something like the microscopic flow speed divided by the pore-throat radius. At the continuum scale, this is given by $\mu/k * u.u$, where μ is viscosity, k is permeability, and $u.u$ is the square of the magnitude of the Darcy flux vector. I'm sure this is in a standard text such as Bear's *Dynamics of Fluids in Porous Media*.

comments by page-line

4-18: Description of the Deborah number unclear. Deform what? viscous relaxation of what?

4-35-40: sentence is way too long and complex

4-43: Methods seems like the wrong section heading name.

4-50: "universally composites" -- awkward phrasing

6-20: "acknowledges a posteriori" -- awkward phrasing. Do you mean a priori?

7-17: "that inclusions have similar properties to surrounding rock" -- this needs a citation and a clearer explanation of what "similar" means.

7-47: See note above; I think your formulation of Darcy dissipation is incorrect.

9-45: Units on the dimensional temperature of 300?

10-7: "expected to be weaker" -- actually I would say that the $(1-\phi)$ factor accounts for the fractional area that is load-bearing.

10-13: "restrict to spherical geometry" -- wasn't this already done above?

16-56: "We will derive an upper..." -- this sentence seems to make no sense (though I think I understand that you are trying to say that the upper bound diverges as the gradient diverges)

18-57: "the smallest data value for permeability of intact bombs..." -- this is an important sentence but the logic is not sufficiently clear. Be explicit about the nature of the "consistency".

(7.2): write this as a multiple of p_{max}^{old} ?

20-38: only a mathematician would use the "representative elementary volume" as a lengthscale "in a volcanological context"! It is a convenient mathematical construct!

Author's Response to Decision Letter for (RSPA-2021-0166.R1)

See Appendix A.

Decision letter (RSPA-2021-0166.R2)

10-Aug-2021

Dear Dr Greenbank

I am pleased to inform you that your manuscript entitled "A theoretical model of Surtseyan bomb fragmentation" has been accepted in its final form for publication in Proceedings A.

Our Production Office will be in contact with you in due course. You can expect to receive a proof of your article soon. Please contact the office to let us know if you are likely to be away from e-mail in the near future. If you do not notify us and comments are not received within 5 days of sending the proof, we may publish the paper as it stands.

As a reminder, you have provided the following 'Data accessibility statement' (if applicable). Please remember to make any data sets live prior to publication, and update any links as needed when you receive a proof to check. It is good practice to also add data sets to your reference list. Statement (if applicable): Data is provided as supplementary material (in the text file supplementarydata.txt). This is original data provided by the third author.

Open access

You are invited to opt for open access, our author pays publishing model. Payment of open access fees will enable your article to be made freely available via the Royal Society website as soon as it is ready for publication. For more information about open access please visit <https://royalsociety.org/journals/authors/which-journal/open-access/>. The open access fee for this journal is £1700/\$2380/€2040 per article. VAT will be charged where applicable.

Note that if you have opted for open access then payment will be required before the article is published – payment instructions will follow shortly.

If you wish to opt for open access then please inform the editorial office (proceedingsa@royalsociety.org) as soon as possible.

Your article has been estimated as being 22 pages long. Our Production Office will inform you of the exact length at the proof stage.

Proceedings A levies charges for articles which exceed 20 printed pages. (based upon approximately 540 words or 2 figures per page). Articles exceeding this limit will incur page charges of £150 per page or part page, plus VAT (where applicable).

Under the terms of our licence to publish you may post the author generated postprint (ie. your accepted version not the final typeset version) of your manuscript at any time and this can be made freely available. Postprints can be deposited on a personal or institutional website, or a recognised server/repository. Please note however, that the reporting of postprints is subject to a media embargo, and that the status the manuscript should be made clear. Upon publication of the definitive version on the publisher's site, full details and a link should be added.

You can cite the article in advance of publication using its DOI. The DOI will take the form: 10.1098/rspa.XXXX.YYYY, where XXXX and YYYY are the last 8 digits of your manuscript number (eg. if your manuscript number is RSPA-2017-1234 the DOI would be 10.1098/rspa.2017.1234).

For tips on promoting your accepted paper see our blog post:
<https://royalsociety.org/blog/2020/07/promoting-your-latest-paper-and-tracking-your-results/>

On behalf of the Editor of Proceedings A, we look forward to your continued contributions to the Journal.

Sincerely,
Raminder Shergill
proceedingsa@royalsociety.org

on behalf of
Professor Colin Meyer
Board Member
Proceedings A

Appendix A

Response to Reviewers RSPA-2021-0166R1

Dr. Emma Greenbank
School of Mathematics and Statistics
Victoria University of Wellington

August 2, 2021

This is in response to referees comments on our manuscript RSPA-2021-0166R1.

We are very happy our manuscript is now accepted subject to minor alterations. We are again very grateful for the time and trouble taken by the referee to provide feedback. We also appreciate the very positive and useful overall comments made.

We are suggesting the cartoon in Fig 1 as a possible cover image for Proc A. We are including a version of the cartoon that is free of numbered arrows for that purpose. The copyright license for this version of the cartoon is a Wikimedia Creative Commons CC-BY-SA3.0 or Free Art License, see the link https://commons.wikimedia.org/wiki/File:Surtseyan_Eruption-blank.svg

We are opting for open access, as our institution (Victoria University of Wellington) has a read-publish agreement with the Royal Society.

We have revised the manuscript as requested and as detailed on the appended pages.

Ngā mihi

Emma Greenbank

Referee comments and our responses:

1. *The writing in the manuscript could be better; there are various places where sentences are convoluted and/or too long, or the wording is awkward or repetitive.*

OK, we've gone through the manuscript with this in mind and tightened phrasing.

2. *Although it is probably inconsequential, their representation of dissipation by Darcy flow is wrong, as I think I mentioned in my previous review. I refer to the unnumbered equation after (3.11), which uses the viscosity times the square of what the authors seem to consider a strain rate. The actual dissipation in porous flow occurs in the Poiseuille-type flow in the pore-throats, at a strain-rate that is something like the microscopic flow speed divided by the pore-throat radius. At the continuum scale, this is given by $\mu/k * u.u$, where μ is viscosity, k is permeability, and $u.u$ is the square of the magnitude of the Darcy flux vector. I'm sure this is in a standard text such as Bear's *Dynamics of Fluids in Porous Media*.*

The calculation of dissipation was based on a microscopic version of the formula. **We have changed it, to be based on the averaged version suggested by the referee, at the continuum scale.** One advantage of this approach is that it allows us to use a maximum pressure difference that corresponds to the critical breaking value (the tensile strength), instead of estimates of vapour velocity. The resulting value of dissipation is another factor of ten smaller than previously calculated, so the outcome is still the same.

Detailed comments by [page:line], where page is the manuscript page number in white on a black box, and line is the

1. *4-18: Description of the Deborah number unclear. Deform what? viscous relaxation of what?*

We have rewritten the paragraph on Deborah number, putting in more math formulae instead of words. We think this explanation is now much clearer.

2. *4-35-40: sentence is way too long and complex*

OK, yes, we've broken it up into three separate sentences.

3. *4-43: Methods seems like the wrong section heading name.*

Yes, we agree. We felt like this was forced upon us by the journal's authors' instructions, that there must be a "methods" section in all papers. We have renamed this section **Field Work and Data**.

4. *4-50: "universally composites" – awkward phrasing*

This phrasing is very concise — the bombs were composites, and there were none

that were not composites — we've made it less concise now though, to read "Bombs were universally observed to be composites"

5. 6-20: *"acknowledges a posteriori" – awkward phrasing. Do you mean a priori?*

That phrasing was borrowed from the other referee. We do mean a posteriori, that is, after solving the model and seeing that initial slurry temperatures had little effect on simulation results. We have now removed the phrase as it will no doubt be awkward for a number of readers, and isn't really needed in this sentence.

6. 7-17: *"that inclusions have similar properties to surrounding rock" – this needs a citation and a clearer explanation of what "similar" means.*

This refers to the Field Work and Data section. We have now inserted a reference to that section.

7. 7-47: *See note above; I think your formulation of Darcy dissipation is incorrect.*

See above; we neglect this term later on, and we have changed the calculations in the magma subsection to better show the averaging results using Darcy's Law for porous media. In the slurry subsection, all terms involving liquid velocity are dropped, including this dissipation term.

8. 9-45: *Units on the dimensional temperature of 300?*

Yes, K now inserted, thanks.

9. 10-7: *"expected to be weaker" – actually I would say that the $(1-\phi)$ factor accounts for the fractional area that is load-bearing.*

Yes, that phrasing is better and we have adopted it, thanks.

10. 10-13: *"restrict to spherical geometry" – wasn't this already done above?*

Oh yes, this is already stated just after equation (3.12), so this reference is now removed from the first sentence of Section 4.

11. 16-56: *"We will derive an upper..." – this sentence seems to make no sense (though I think I understand that you are trying to say that the upper bound diverges as the gradient diverges)*

OK we have rephrased to read "We now derive an upper bound on the maximum pressure. We find that our bound diverges as the initial temperature gradient diverges, suggesting that the pressure maximum may be theoretically unbounded in this limit. "

12. 18-57: *"the smallest data value for permeability of intact bombs..." – this is an important sentence but the logic is not sufficiently clear. Be explicit about the nature*

of the "consistency".

OK this explanation/discussion has been expanded and altered for better clarity.

13. (7.2): write this as a multiple of p_{max}^{old} ?

Yes, good idea, done.

14. 20-38: only a mathematician would use the "representative elementary volume" as a lengthscale "in a volcanological context"! It is a convenient mathematical construct!

Ha ha, now I could imagine that a volcanologist might have a very clear idea of the size of an REV, small enough to fit many of them into a single bomb, but large enough that the averaging of material properties across some number of pores and grains is justified. I would also hope that the REV is in fact a *modelling* construct, of equal value and convenience to both volcanologists and mathematicians! That said, I'm aware of an increasing awareness in nanotechnology and fluid mechanics that the continuum approximation gives good matches to data even when used over distances the size of a water molecule.